# Center-surround interactions underlie bipolar cell motion sensitivity in the mouse retina

Sarah Strauss[1,2,3,6], Maria M. Korympidou[1,2,6], Yanli Ran[1,2], Katrin Franke [1,2], Timm Schubert [1,2], Tom Baden [1,4], Philipp Berens [1,2,3], Thomas Euler [1,2] ✉ & Anna L. Vlasits [1,2,5] ✉

Motion sensing is a critical aspect of vision. We studied the representation of motion in mouse retinal bipolar cells and found that some bipolar cells are radially direction selective, preferring the origin of small object motion trajectories. Using a glutamate sensor, we directly observed bipolar cells synaptic output and found that there are radial direction selective and non-selective bipolar cell types, the majority being selective, and that radial direction selectivity relies on properties of the center-surround receptive field. We used these bipolar cell receptive fields along with connectomics to design biophysical models of downstream cells. The models and additional experiments demonstrated that bipolar cells pass radial direction selective excitation to starburst amacrine cells, which contributes to their directional tuning. As bipolar cells provide excitation to most amacrine and ganglion cells, their radial direction selectivity may contribute to motion processing throughout the visual system.

Local motion sensing is of paramount importance for sighted animals, enabling them to detect and capture prey[1–4], as well as to avoid predators[5–9]. In mammals, multiple features related to motion sensing are first extracted from the visual scene by the retina[10,11]. These features include the direction of motion[12,13], looming motion[14], and differential motion[15], and can be used, for instance, to filter the local motion of objects from the global motion caused by body, head, and eye movements. The stages at which motion is extracted in the retinal circuitry and the mechanisms of motion-related feature detection are key to understanding these processes.

Motion features are most likely to be computed in the inner retina. There, 14 types of bipolar cells (BCs,[16–20]) receive input from photoreceptors. BCs provide excitatory glutamatergic input to a large diversity of amacrine cells (ACs), which are a class of inhibitory interneurons (reviewed in[21]), and retinal ganglion cells (RGCs), which are the output neurons of the retina (reviewed in refs. 22,23). Although BCs represent the first stage in the retina where visual signals diverge into parallel channels, motion

detection has not yet been found to be widely implemented at the BC level.

For instance, one well-studied motion detection circuit in the retina is the direction selectivity (DS) circuit, where the BCs' role in the motion computation remains intensely debated. One key element in this DS circuit is the starburst amacrine cell (SAC), which exhibits DS for motion at the level of individual neurites[24], providing asymmetric inhibition to direction selective RGCs during motion in one direction[25–29]. The role of BCs in this DS circuit has been a matter of intense scrutiny, with a variety of studies providing evidence supporting an important role for BCs in the motion computation by SACs (refs. 19, 30, 31, but also see refs. 32, 33), or by direction selective RGCs[34,35]. More specifically, it was suggested that distinct BC types with different glutamate release kinetics[16,36,37] provide spatially offset inputs on postsynaptic SAC dendrites ("space-time" wiring,[30]), enhancing the preferred direction response. While voltage-clamp recordings in the rabbit retina indicate some directional tuning in BCs[38–40], observations of BC Ca²⁺ and glutamate release suggest that BCs respond

[1]Institute for Ophthalmic Research, University of Tübingen, Tübingen, Germany. [2]Centre for Integrative Neuroscience, University of Tübingen, Tübingen, Germany. [3]Tübingen AI Center, University of Tübingen, Tübingen, Germany. [4]School of Life Sciences, University of Sussex, Brighton, UK. [5]Present address: Department of Neurobiology, Northwestern University, Evanston, IL, USA. [6]These authors contributed equally: Sarah Strauss, Maria M. Korympidou. ✉e-mail: thomas.euler@cin.uni-tuebingen.de; anna.vlasits@northwestern.edu

symmetrically to motion stimuli[41–44] (but see ref. [45]). Therefore, BCs have not been considered direction-tuned cells themselves and their exact role in the DS circuit remains debated.

At the same time, BCs exhibit a receptive field (RF) feature that could support another basic form of motion detection: their center-surround antagonism[46–49]. A phenomenon of many sensory neurons, center-surround antagonism refers to a preference for opposite polarity stimuli in the center of the RF vs. the surround. For instance, so-called On BCs depolarize and release more glutamate to light increments in the center (a light turning "On"), while the surround antagonistically inhibits the center to light increments. Off BCs have the opposite preference. Antagonistic interactions between RF center and surround enrich the BC types' functional diversity, as they possess differences in the size and strength of center and surround as well as in the temporal relationship between center and surround responses[16,50]. These interactions are partially established by horizontal cells in the outer plexiform layer (reviewed in ref. [51]), but importantly shaped further by ACs[16,52]. More than 50 years ago, it was hypothesized that the interplay between spatially and temporally offset excitation and inhibition establishes retinal motion detectors[53]. Yet, the role of BCs' antagonistic center-surround interactions in motion detection has received little attention[54–56]. Specifically, whether BCs respond differently to motion originating in their center vs. in their surround ("radial direction selectivity", rDS) has not been explicitly explored.

Here, we studied the local motion sensing properties of BCs throughout the inner retina by measuring BC output using a fluorescent glutamate sensor during visual stimulation. We found that some BCs exhibit a sensitivity to the origin of a motion trajectory–rDS with a preference for centrifugal, outward motion. To explore this further, we characterized the center-surround RFs of BCs across the inner plexiform layer (IPL) and uncovered diversity in their RF properties for motion sensing, which confers rDS and looming sensitivity to a subpopulation of BC types. We explored the implications of this rDS for downstream retinal processing by constructing biophysical models of SAC dendrites with anatomically and spatio-temporally precise input from BCs. We found that the SAC inherits motion signals from BCs and their DS is diminished by *in-silico* removal of the BCs' RF surrounds. Last, we verified our *in-silico* findings experimentally by measuring Ca²⁺ dynamics in SAC dendrites. Our findings suggest that BCs produce radial direction selective signals, and that these signals can play a role in the computation of local motion direction in SACs. Given the central role of BCs in retinal signaling, our findings suggest that BCs may play a key role in many motion computations throughout the retina.

## Results

### Bipolar cell glutamate release is sensitive to motion origin

To observe how BCs respond to small, locally moving stimuli, we performed 2-photon imaging of a glutamate sensor, iGluSnFR[16,44] expressed throughout the neurons of the IPL (Fig. 1a). We began by imaging at relatively low spatial resolution to observe glutamate release dynamics in the On layer of the IPL over a large field of view (FOV, ~200 µm²) during visual motion stimulation (Fig. 1b). We presented a small, bright moving bar (20 × 40 µm) that traversed a distance of 100 µm, corresponding to roughly 3.3° of visual angle[57] spanning the width of 2–4 BC RFs, in two opposite directions. Glutamate signals from this stimulus were complex, with changes in glutamate release occurring throughout the FOV, well beyond the bounds of the stimulus and its trajectory. We observed that the response amplitude in some regions of the FOV depended on the direction of stimuli. Specifically, BCs in areas where motion originated and terminated appeared to release more glutamate when motion originated nearby. In other regions, the glutamate release appeared symmetric between motion directions.

Based on these observations, we sought to measure whether individual BC terminals exhibit different responses to objects at different points on their motion trajectories. We performed iGluSnFR imaging at higher spatial resolution in the On layer of the IPL (Fig. 1d) and presented moving stimuli originating inside the FOV and moving out ("originates", green) or outside of the FOV and moving in ("terminates", cyan) or passing through the FOV in one of two directions ("passes through", black) (Fig. 1e). To better capture the activity of small, noisy regions of interest (ROIs) that were the size of BC terminals[16] (see Methods for details), we used Gaussian Process modeling to infer the mean and standard deviation (s.d.) of individual ROI responses to each stimulus condition[58] (Supplementary Fig. 1). We observed that many ROIs exhibit strong glutamate release to motion originating in the FOV, and that ROIs preferred this stimulus to motion that terminated in the FOV or passed through, which we termed "radial direction selectivity" (rDS) (Fig. 1f). We calculated the extent of this preference (d-prime, d') to examine stimulus preference across the FOV (Fig. 1g, h). We found that the preference for originating motion was restricted to a small region of about the size of a BC's RF center, and that no such preference existed between stimuli passing through in different directions (Fig. 1h, data from $n = 4056$ ROIs/ 7 fields/ 3 mice). This implies that BCs signal the location of motion origin with high spatial precision. In addition, we found that within the area of the FOV where the motion originated, the d' across ROIs was significantly shifted toward positive values, signifying rDS (Fig. 1i, j, 100 µm, $d' = 41.0 \pm 54.1$; 150 µm, $d' = -2.4 \pm 26.7$; 300 µm, $d' = 4.4 \pm 30.0$; $p < 0.01$, Wilcoxon signed-rank test, $n = 641$ ROIs/ 7 fields/ 3 mice). These results suggest that at least some BCs are highly sensitive to motion radiating out from their RF centers.

### Bipolar cells exhibit differing sensitivity to motion

Initial observations of responses to moving bar stimuli suggested that not all BCs possess rDS (Supplementary Fig. 2). Previous measures of BC response properties suggest that the 14 BC types differ in their spatial and temporal response properties and kinetics[16,37,59,60]. These differences could be important for rDS. In order to understand the relationship between RF properties and rDS of BCs, we mapped BC RFs and measured the rDS of those RFs. We used 2-photon imaging enabled by an electrically-tunable lens[61]. This allowed "axial" (x–z) scans and, hence, imaging glutamate release across all IPL layers in a single imaging frame. We measured the RF properties of BC glutamate release using a "1D noise" stimulus, which captures the temporal as well as one spatial dimension of the RF, (Fig. 2a) and inferred smooth RFs using a spline-based RF estimation method[62]. In this way, we could observe center-surround RFs from ROIs near the size of individual BC boutons (ROI sizes ~2 µm², see "Methods"), including clear On and Off RFs from their respective IPL strata (Fig. 2b) for 3233 ROIs. We clustered these RFs into groups using a Mixture of Gaussian (MoG) clustering on features from the RFs as well as each ROI's IPL depth, and uncovered 13 clusters of BC RFs (Fig. 2c–e). Individual clusters contained ROIs stratifying tightly in the IPL (Fig. 2f) and most clusters exhibited stereotyped temporal properties of their centers and surrounds within cluster (Supplementary Fig. 3). We computed the average RF for each cluster and observed that these average RFs had distinct properties, most notably the temporal and spatial characteristics of the surround (Fig. 2g, h, see also Fig. 3).

To evaluate the rDS of individual BC clusters, we modeled their responses to a moving bar stimulus by convolving the average RF for each cluster with a space-time stimulus image. We cross-validated this method of modeling by comparing individual ROIs' real moving bar responses (as in Fig. 1) to model predictions of motion responses in imaging fields presented with both the moving bar and noise stimuli. We found that the linear convolution model predicts moving bar responses with good accuracy (Supplementary Fig. 4, median Pearson correlation for originating motion = 0.61, $n = 878$ ROIs). To test the

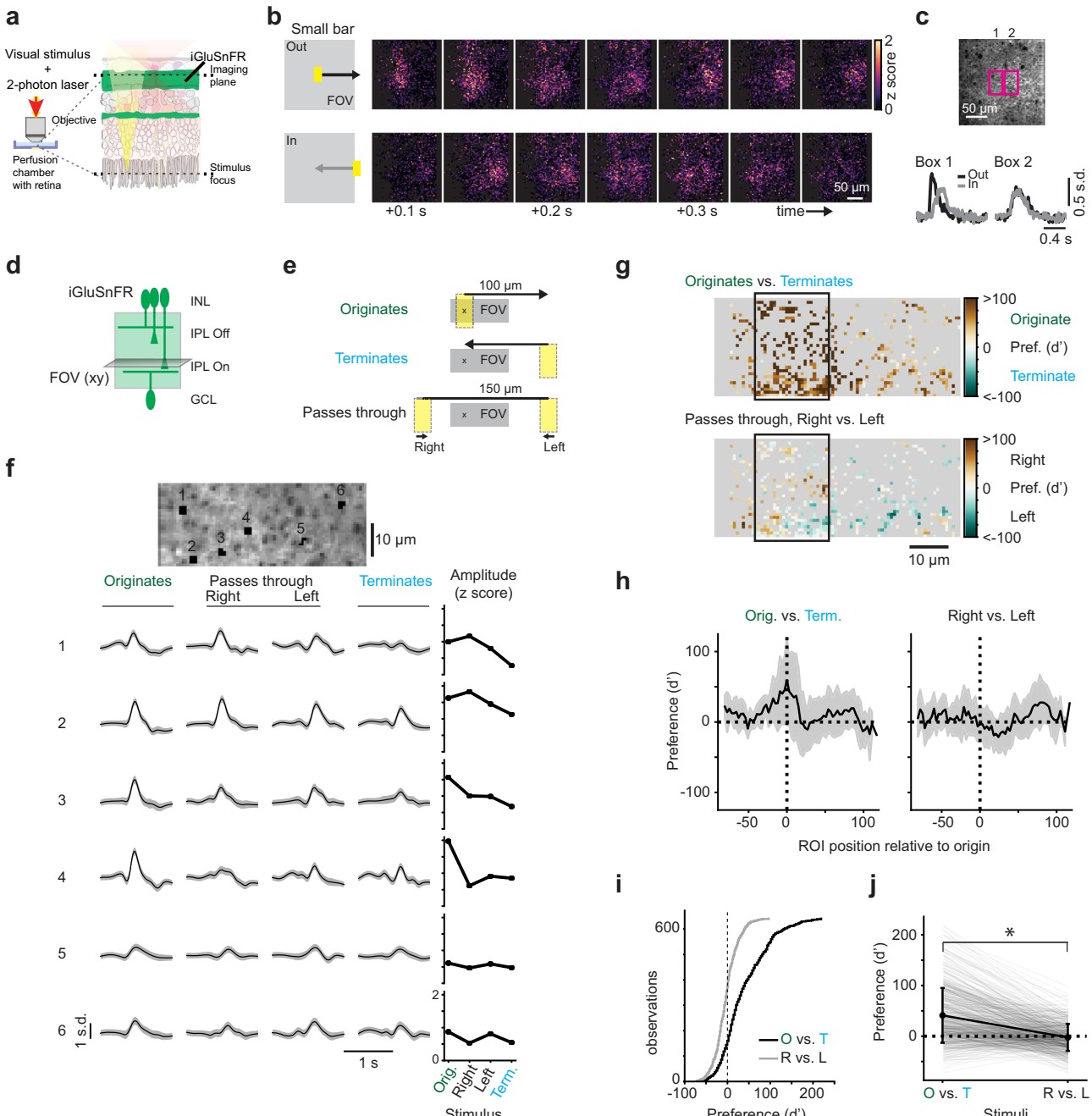

**Fig. 1 | Bipolar cell glutamate release is radially direction selective.**
**a** Experimental setup showing objective, retina, visual stimulus (yellow), and 2-photon laser (red). Adapted from ref. 61. **b** Left: stimulus ($20 \times 40\,\mu m$ light rectangle on dark background) moving at $500\,\mu m/s$ over $100\,\mu m$, starting either in the field of view (FOV) center or just outside the FOV. Right: montage of the average z-scored fluorescence of glutamate sensor iGluSnFR during stimulation. Representative example, experiment repeated in 3 retinas/2 animals. **c** Top: average iGluSnFR fluorescence showing two regions of interest (ROIs). Bottom, mean binned fluorescence for the pixels in each ROI. **d** iGluSnFR is ubiquitously expressed in retinal neurons, including in the cells of the inner plexiform layer (IPL, green region). INL, inner nuclear layer; GCL, ganglion cell layer. **e** Moving bars ($20 \times 40\,\mu m$) presented to the retina originating in the FOV (green), terminating in the FOV (cyan), or passing through in two directions (black). All objects to scale. **f** Example ROIs (black regions, numbered) overlaid with s.d. of the imaged field and

their responses to the stimuli in (b) as predicted using Gaussian Process modeling. Black, estimated mean; gray shading, 3 s.d. Rightmost column: maximum response amplitude for each stimulus condition. **g** Motion preference (d') for all ROIs in example field. The boxed region is the starting position of the "originates" condition and is analyzed in (i) and (j). **h** d' for originating vs. terminating or passing through stimuli as a function of location relative to the start position of originating motion ("x" in e). Black: mean, gray band: s.d. Sample size is $n = 4056$ ROIs/ 7 fields/ 3 mice. **i** Cumulative histogram of d' for ROIs located within $10\,\mu m$ on either side of the "originates" stimulus start position ("x" in e; black rectangle in g). **j** d' of each ROI in the population. Black, mean values ± s.d.; gray, individual ROIs. These conditions are significantly different ($p = 1.53e - 61$, two-sided Wilcoxon signed-rank test). Sample size in (**i**, **j**) is 641 ROIs/ 7 fields/ 3 mice. See also Supplementary Figs. 1 and 2.

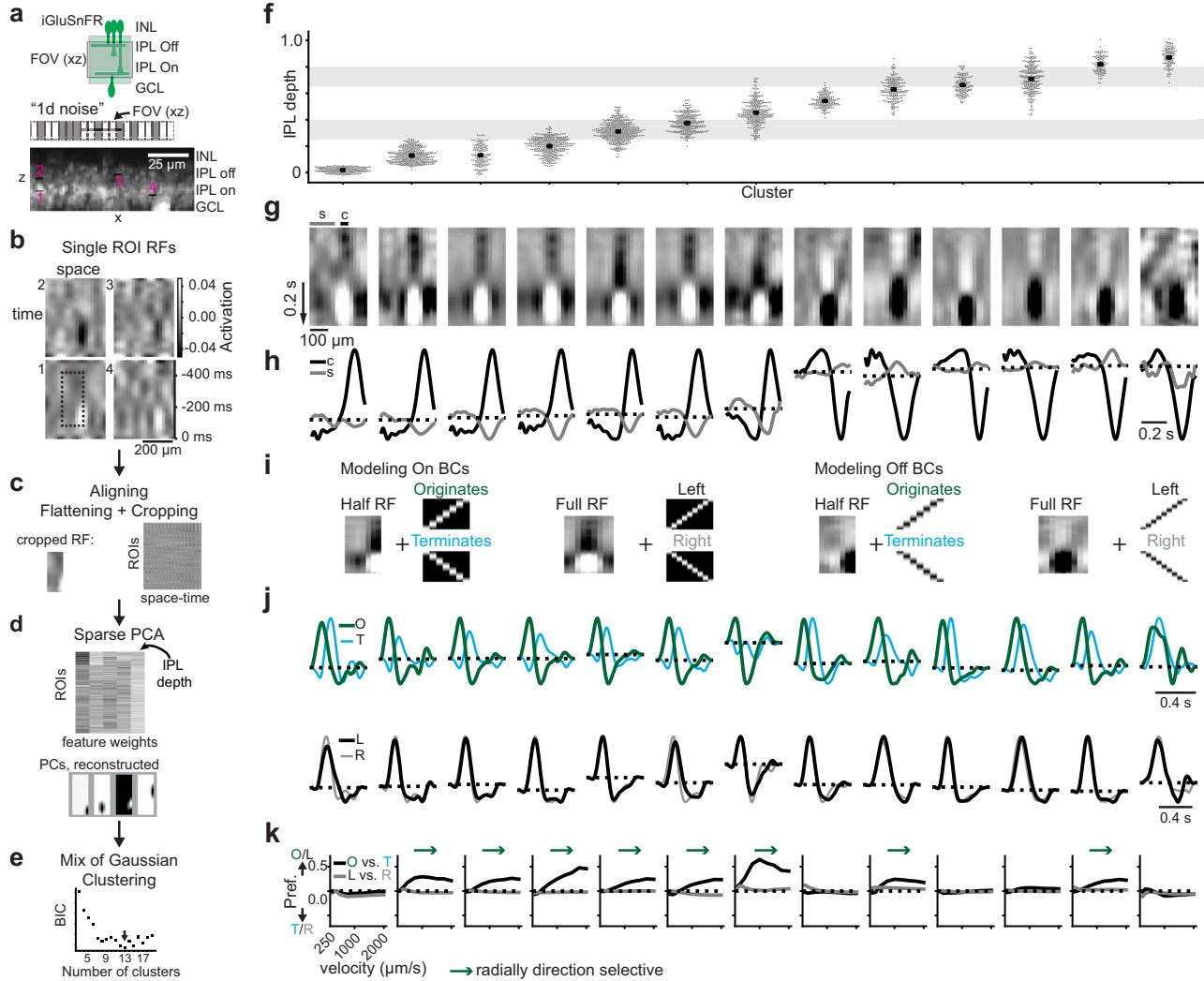

**Fig. 2 | Bipolar cell receptive fields exhibit differing radial direction selectivity.** **a** Top: iGluSnFR is ubiquitously expressed in the IPL, which was vertically scanned using an electrically-tunable lens. Middle: "1D noise" stimulus consisting of twenty 20 × 50 μm rectangles switching randomly between black and white at 20 Hz. The relative scale of the FOV is shown. Bottom: average of a scan with example regions of interest (ROIs, black regions in **b**). **b** Example RFs from the ROIs in (**a**). Dotted box: the cropped RF used for clustering. **c** Top: the cropped RF from the dotted box in (**b**), with the same aspect ratio as the principal components (PCs) in (**d**). Bottom: RFs were aligned to the RF center and then all RFs were flattened to 1 dimension and cropped to exclude missing space-time. Sample size is 3,233 ROIs/ 4 fields/ 3 mice. **d** Top: feature weights for the 4 components from PCA. The fifth feature was the IPL depth of the ROI. Bottom, reconstructed components of the sparse PCA. **e** Mixture of Gaussian clustering was performed and the Bayesian information criterion (BIC)

was used to select the number of clusters. **f** Clusters plotted against IPL depth. Gray regions: approximate ChAT bands as IPL landmarks. **g** Average RF of each cluster. "c" and "s" show the regions used to calculate the spatial average of the center and surround in (**h**). **h** Average temporal RFs taken from center ("c") and surround ("s") regions indicated in (**g**), normalized to the peak of the center response. **i** Example cropped RFs ("half" vs. "full") convolved with the motion stimuli ("originates", "terminates", "left", "right") to measure the rDS of each cluster. **j** Modeled responses to motion (velocity 1000 μm/s) for each cluster for stimuli originating ("O", green) or terminating ("T", cyan) in the RF center or passing through the RF ("L", black; "R", gray). **k** Stimulus preference as a function of velocity for each cluster for the originating vs. terminating (black) or passing through to the left or right (gray) conditions. See also Supplementary Figs. 3–6.

radial direction preference of BC RFs, we modeled the response to a moving stimulus originating in the RF center ("originates", green) or terminating in the RF center ("terminates", cyan) (Fig. 2i). We compared this scenario to the case of a bar moving through the RF, from surround to center and then surround again in one of two directions ("left" vs. "right"). We found that some BC clusters exhibited a preference for motion originating in their RF centers, while others showed no preference for originating vs. terminating motion (Fig. 2j, k). We modeled these responses across a range of stimulus velocities and measured rDS across velocities. Our modeling revealed that some BC clusters in both On and Off layers exhibited rDS across a range of velocities, while other clusters had no direction preference (Fig. 2k). A similar diversity of rDS strength was obtained from an alternative clustering that excluded anatomical IPL depth as a feature for

clustering (Supplementary Fig. 5). We also found that rDS was not limited to the moving bar stimulus, but that motion-sensitive BCs also preferred looming stimuli to receding stimuli (Supplementary Fig. 6). This suggests that specific BC types might be important for several types of motion sensing tasks that are known to be performed within the retina[4,6,10].

## Layer-specific radial direction selectivity depends on bipolar cell surround

To determine which features of the BC RFs are important to establish rDS, we measured several properties of the cluster RFs and found that longer center-surround latency and stronger surround strength were correlated with increased rDS (Fig. 3a, center-surround latency vs. rDS preference, Spearman correlation $\rho = -0.89$, $p < 0.01$; surround

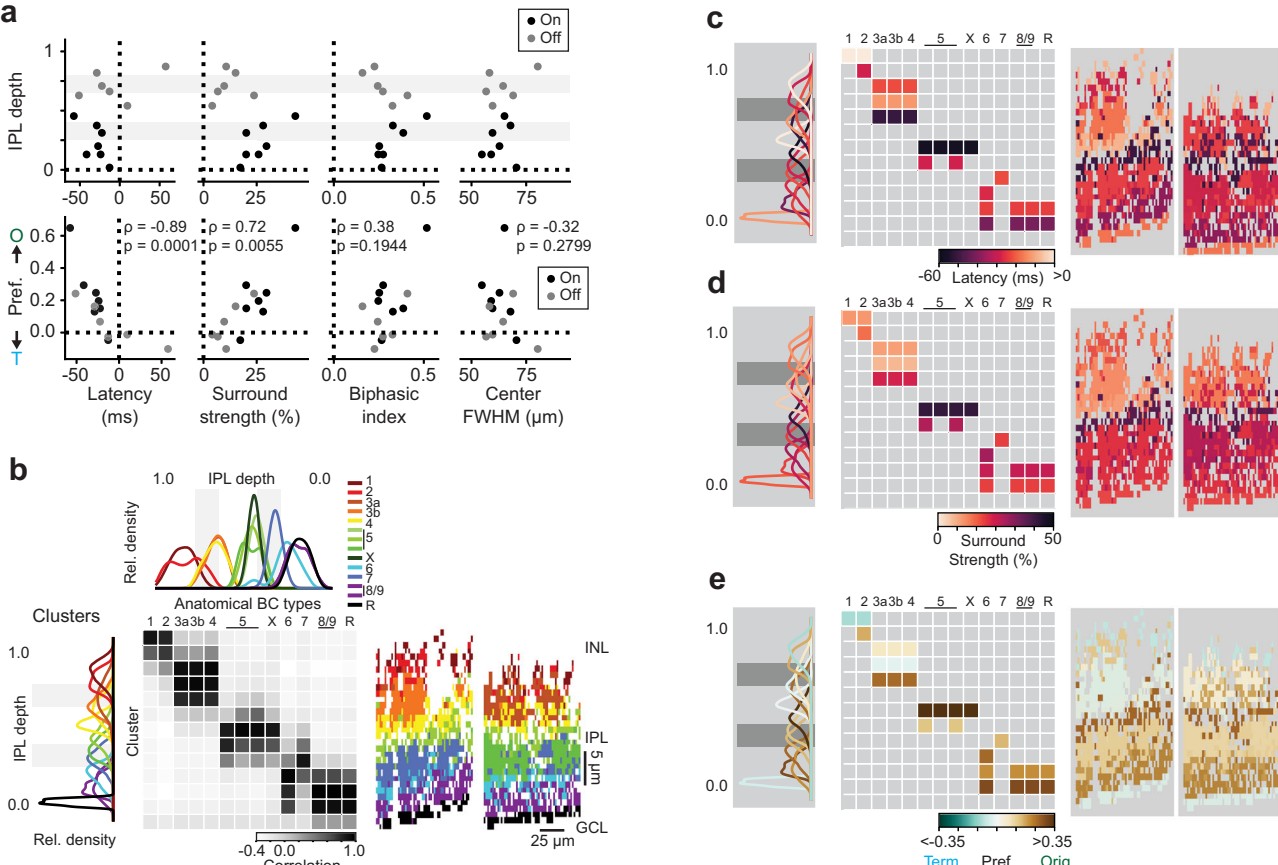

**Fig. 3 | Layer-specific radial direction selectivity depends on bipolar cell surround. a** Center-surround properties of BC clusters from Fig. 2 plotted against IPL depth (top) or rDS preference of modeled responses (bottom). Black and gray points represent On and Off-type BCs, respectively; gray shading marks approx. ChAT bands. $\rho$ indicates the Spearman correlation coefficient, with p-values listed (two-sided test). **b** Top: density plot of BC anatomical stratification as reported earlier[18,19,30]. Left: density plot of BC cluster stratification of ROIs from each cluster in Fig. 2. Colors chosen by likely matches with anatomical stratification. Gray shading marks approx. ChAT bands. Middle: correlation between BC clusters and anatomical types based on their stratification in the IPL. Right: cluster assignment and likely BC type mapped onto pixels from two example imaging fields. **c** Left: density plot from (**b**) color coded by the latency between peak of surround and center responses. Middle: cluster latency assigned to squares with greater than 0.7 correlation based on IPL stratification in (**b**). Right: cluster latency of surround vs. center mapped onto ROIs from two imaging fields (same fields as (**b**)). **d** Same as (**c**), but with all panels color coded by the cluster surround strength relative to center strength. **e** Same as (**c**), but with all panels color coded by the cluster rDS measured from RF convolution with 1000 μm/s velocity motion stimulus (Fig. 2j).

strength vs. origin preference, $\rho = 0.72$, $p < 0.01$), while properties of the center were not (biphasic index vs. rDS preference, $\rho = 0.38$, $p = 0.19$; center full-width half-max (FWHM) vs. rDS preference, $\rho = -0.32$, $p = 0.28$). We confirmed the critical role of the BC RFs' surround in the modeled rDS preferences by decreasing the strength of the surround artificially (Supplementary Fig. 7). Then, by decomposing the modeled responses into contributions from the center and surround, we observed that the inhibition from the surround is more temporally-offset from the excitatory center during outward motion compared to inward motion (Supplementary Fig. 7).

We next asked how the BC clusters grouped by their RF features correspond to known anatomical BC types. We compared the distribution of IPL depths for ROIs within each cluster to the distribution of BC axonal stratification for types identified from electron microscopy (EM, data from refs. 18, 19, 30) and found that the number and extent of co-stratifying clusters was correlated with the number and extent of anatomical BC types (Fig. 3b). For instance, we observed three clusters co-stratifying with the stratification band for the three BC types 3a, 3b, and 4. In addition, some of our BC clusters showed a strong correlation with single EM clusters (type 6, type 7). Thus, we argue that these clusters represent distinct types of BCs.

Next, we explored how RF features and rDS map onto IPL stratification and anatomical type (Fig. 3c–e). Within groups of co-stratifying BC types, we found a diversity of RF properties and rDS. Notably, we observed that at least one type within each sublamina of the IPL exhibited rDS (Fig. 3e), suggesting that this functional response property is accessible to post-synaptic partners throughout the IPL.

## Bipolar cells prefer motion to non-motion stimuli

Our results indicate that the relative strength and timing of the surround vs. the center are correlated with stronger rDS in BCs. We propose that the BC center-surround RF acts as a Barlow-Levick motion detector[53], in which spatial and temporal offset of the inhibitory surround and excitatory center establish selectivity for outward radial motion (Fig. 4a). Given this property, we wondered whether BCs would have a more general preference for motion compared to non-motion stimuli, a preference previously observed in downstream neurons in the retina using stimuli that compare sequential presentation of adjacent bars (apparent "motion") to the same bars presented in random order ("random", refs. 54, 63, Fig. 4c). We imaged glutamate release in the IPL in response to these stimuli and characterized the same ROIs' RFs using the "1-d noise" stimulus (Fig. 4b, d).

First, we modeled responses to motion vs. random stimuli using RFs obtained from noise stimulation and our linear convolution approach (Fig. 4e). In all On BC clusters, we found that responses to apparent motion were stronger than responses to random sequences

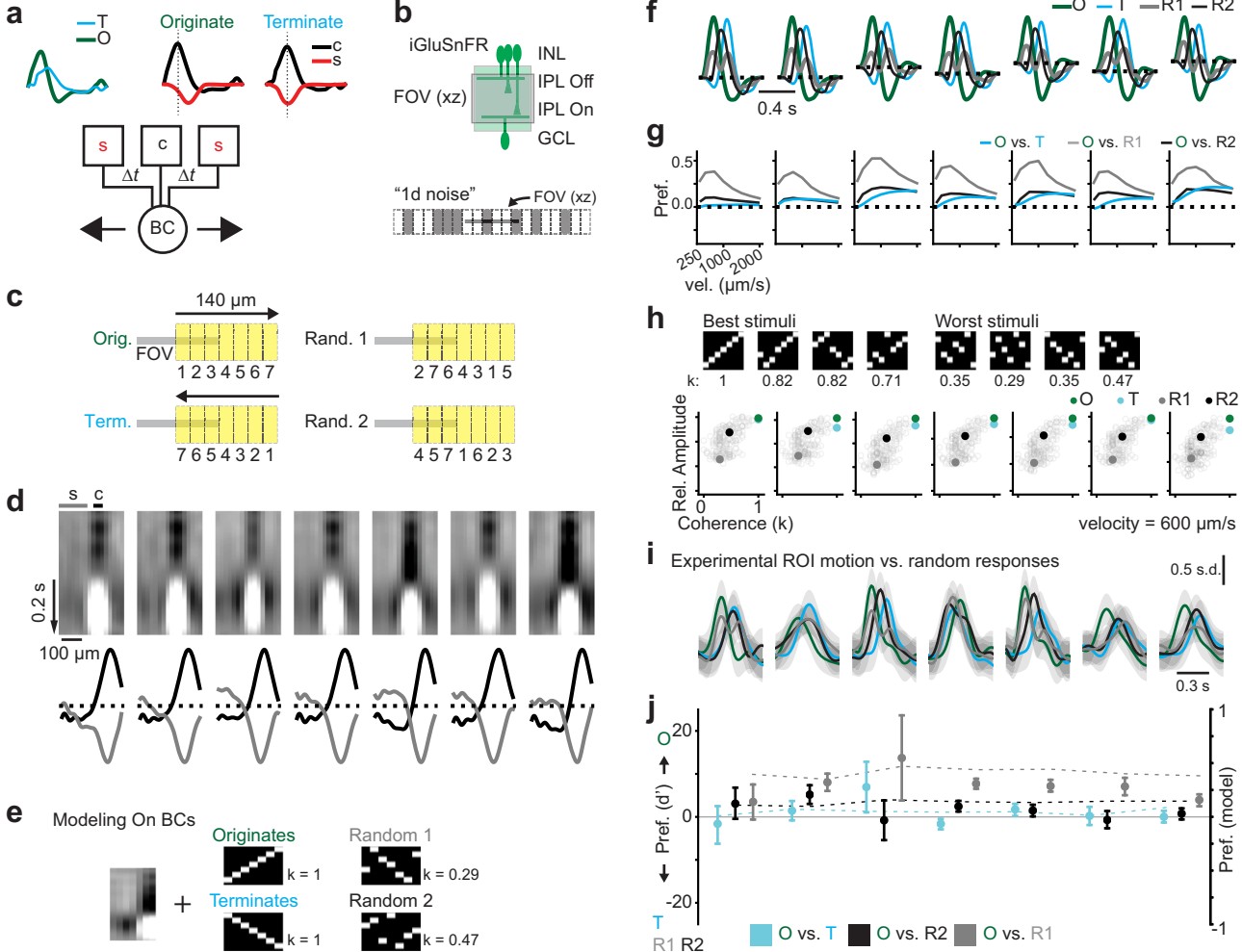

**Fig. 4 | Bipolar cells prefer motion to non-motion stimuli. a** Modeled response of one BC cluster (see Supplementary Fig. 10) to outward and inward motion. Responses were decomposed into BC RF center (**c**, black) and surround (**s**, red) contributions (Supplementary Fig. 7). These coincide during inward motion, leading to a smaller activation according to a Barlow-Levick detector (bottom)[53]. Δt, temporal offset. **b** iGluSnFR imaging and "1D noise" stimulus to assign cluster membership and predict motion responses. **c** Stimuli used to assess stimulus preference, by presenting seven 20 × 50 μm rectangles in sequences to create apparent originating ("Orig."), terminating ("Term.") motion, or one of two random ("Rand.") sequences. Apparent velocity, 600 μm/s. **d** Top: average RF of each cluster. "c" and "s" show the regions used to calculate the spatial average of the center and surround below. Bottom: average temporal RFs taken from "c" and "s" regions indicated in (**d**), normalized to their respective peaks. Sample size is 5837 ROIs/ 5 fields/ 3 mice. **e** Example RF convolved with the motion and random stimuli

to model the motion responses of each cluster. k: stimulus motion coherence. **f** Linear convolution model response predictions (velocity, 600 μm/s) for originating ("O"), terminating ("T"), and random ("R") motion. **g** Preference for originating motion compared to terminating motion or random sequences as a function of velocity. **h** Relative maximum amplitude of the modeled responses to 200 random stimuli as a function of k for each cluster (gray open circles). Filled circles show the sequences tested experimentally. Top insets show example stimuli that caused the largest ("best") vs. smallest ("worst") amplitude responses with their k listed below. **i** Example experimental responses of single ROIs from each cluster to stimuli in (**c**). Gaussian Process predictions as in Fig. 1. Lines, estimated means; gray shading, 3 s.d. **j** Preference (d') to stimuli in (**c**) for ROIs in each On BC cluster. Data points, the mean; error bars, 95% confidence interval. Sample includes 1801 ROIs/ 5 fields/ 3 mice. Dotted lines: preferences predicted from the modeled responses in (**f**).

(Fig. 4f). This preference was most pronounced at velocities below 1000 μm/s (Fig. 4g). Interestingly, we observed a difference in the predicted responses to the two random sequences. To understand the relationship between the stimulus sequences and responses, we simulated an additional 200 random sequences. We observed that stimuli that produced large amplitude responses tended to contain more coherent motion than stimuli producing small amplitude responses (Fig. 4h). We calculated a measure of motion coherence (see Methods) and examined the relationship with the modeled response amplitude. We found that there is a strong positive relationship between motion coherence and predicted response amplitude.

We confirmed these predictions experimentally by examining single ROI responses to motion vs. random stimuli (Fig. 4i). We found that our model predictions matched the experimental responses, with a preference for coherent motion stimuli compared to the least

coherent stimulus at the velocity we tested (~600 μm/s; 2-way repeated measures ANOVA: $p < 0.01$ for the effect of stimulus, but not cluster or their interaction; Bonferroni post-hoc comparisons: $p < 0.01$ for each stimulus comparison) (Fig. 4j). Together, these results suggest that one important role for BCs is the separation of motion vs. non-motion signals, a phenomenon observed throughout the mammalian visual system (i.e., in mouse[64–66]).

## Radial direction selectivity is shaped by GABAergic signaling in some bipolar cell clusters

BC motion-related surround properties might be shaped at several stages in the retinal circuit via feedback and feedforward signaling from horizontal cells and amacrine cells. Given the diversity of surround features we observed in our RF mapping, we hypothesized that rDS is shaped in a type-specific manner by AC inputs. We tested the

effects of two blockers of inhibition in the retina, (1,2,5,6-Tetra-hydropyridin-4-yl)methylphosphinic acid (TPMPA) to block GABA$_C$ receptors, which are commonly found on BCs[67,68], and strychnine to block glycinergic signaling, which have been shown to change BC RF properties[16]. We measured BC glutamate release during stimulation with both noise and moving bars (Fig. 5a) to obtain RFs and responses to motion. We used the RFs obtained from noise in control conditions to assign ROIs to the clusters identified in Fig. 2. We observed several effects of TPMPA on the cluster average RFs, including changes in the strength of the surround (Fig. 5b, e) and widening of the center (Fig. 5c), a phenomenon that was previously observed[16]. We modeled responses to originating and terminating motion and observed that several clusters exhibited large changes in their rDS across velocities in control compared to drug conditions (Fig. 5d). The changes were correlated with changes in surround strength (Fig. 5e). Next, we examined the responses to moving bars and found that two clusters that our simulations had predicted to lose their rDS indeed exhibited reduced rDS in TPMPA. A similar analysis of the strychnine data revealed no effect on rDS, although some RF features were altered (see Supplementary Fig. 8). Thus, GABAergic signaling mediated by GABA$_C$ receptors appears to boost rDS of some bipolar cell clusters but not others.

## Starburst amacrine cells inherit bipolar cell motion information

Given the presence of radially direction selective BCs throughout the IPL, we wondered whether BCs' post-synaptic partners use this information for motion-related computations. To explore this issue, we studied whether SACs in the direction selective circuit, which were shown to display a preference for motion from their somas to their distal dendrites[24,69], inherit motion information from BCs. First, we confirmed that BC inputs onto SACs are radially direction selective by imaging glutamate release onto On layer SAC dendrites through targeted expression of flex-iGluSnFR under control of the ChAT promoter (Fig. 6a). We measured iGluSnFR signals in response to moving bar (Supplementary Fig. 9) and noise stimuli (Supplementary Fig. 10). We observed a similar pattern of rDS as we described for iGluSnFR expressed throughout the IPL (Fig. 6b). In addition, we observed a preference for looming motion compared to receding motion in SAC-layer glutamate release (Supplementary Fig. 6). Finally, we measured and clustered the RFs of BCs releasing glutamate onto the dendrites of On layer SACs and uncovered five clusters of RFs, which exhibited a mixture of radial direction selective and non-selective types predicted from linear convolution models (Fig. 6c). These clusters' velocity tuning covered a similar range of rDS as the tuning of On BC clusters revealed in Fig. 2. All together, these results indicate that BC glutamate release onto SACs is radially direction selective, and that in some stimulus conditions, this asymmetric glutamate release could contribute to motion computations in this AC type.

Next, we explored how BC motion information is integrated by SACs by constructing biophysical models of On and Off SAC dendrites. Our models were based on previous SAC models and used published connectomic and physiological data about the BC types and locations of BC inputs on SAC dendrites[32,57]. We omitted inhibitory connections to SACs in order to highlight the specific role played by BCs. Where previous models included none of the center-surround dynamics of the BC RFs, we modeled these spatio-temporal dynamics using RFs derived from specific BC clusters (Fig. 2). We selected cluster RFs that were likely matches with BC types known to provide input to SACs (Fig. 6d, e) (see Supplementary Figs. 7 and 10 for details). We observed that many of the BC types chosen for model input were radially direction selective in our simulations.

To examine the direction selectivity of our SAC models, we simulated a moving bar stimulus that traversed the length of the dendrite in the centrifugal (from soma to distal dendrite, CF) or centripetal (from distal dendrite to soma, CP) direction. We monitored the

voltage along the entire model dendrite and found that the model responded with asymmetric depolarization with a preference for CF motion (Fig. 6f). In distal model compartments, where SACs have their output synapses, the difference between CF and CP motion was particularly pronounced, and we observed DS across a wide range of physiologically- and behaviorally-relevant velocities (Fig. 6g)[3,4,64,66,70,71]. Thus, the anatomical wiring between BCs and the SAC dendrite, together with the BC RF properties, enable stronger activation of BCs along the SAC dendrite during CF motion compared to CP motion in our model (Fig. 6h). Previous research on connectomic reconstructions of the SAC has suggested that the gradient of BC types along the SAC dendrite plays a role in their DS[19,30,31]. We explored this issue by changing which functional RF clusters provide input to our models. We found that some BC cluster RFs led to stronger DS in the SAC model, while others produced weaker DS (Fig. 6i and Supplementary Fig. 10). Our results suggest that the Off SAC model is more sensitive to changes in the BC distribution, as all alternative distributions resulted in a reduction of the DS compared to the original one. In the On SAC model, on the other hand, the modifications of the BC distributions resulted in differential changes in the strength of the direction preference (direction selectivity index, DSI). These results reflect the differences in the BC RF properties and anatomical wiring between the On and Off pathways[19,30,57]. Thus, BC RFs appear to contribute to SAC DS, and this contribution depends on BC input identity and RF properties.

Next, we explored the influence of the BC RF properties on the DS of our SAC models by testing variations in BC surround strength and different spatial stimulation. We tested both weaker and stronger surrounds, particularly because our method of obtaining RFs likely underestimates surround strength (see ref. 72 and "Methods") and because surround strength can be dynamically altered by environmental conditions[73]. First, we changed the strength of the surround component of the BC cluster RFs (Supplementary Fig. 7). We observed that the model responses to motion were strongly influenced by surround strength, especially in the CP motion direction, presenting a marked surround strength dependence of DS tuning across stimulus velocities (Fig. 7a–c). In addition, we found that the RF surround was the dominant feature conferring DS to the SAC models (Supplementary Fig. 10). We then evaluated how this surround dependence affects SAC model responses to stimuli traversing different spatial locations and distances relative to the dendrite. In particular, we tested if stimulating the proximal BC RF inputs more symmetrically using stimuli that pass through rather than originate over their RF centers (Fig. 1) produce less DS in the SAC ("Cell diameter"). Also, we tested if stimuli that activate more of the surround of the distal BC inputs ("Cell surround") produce stronger DS (Fig. 7d). We found that spatial stimulation indeed produced these effects in the model SAC dendrites, especially at high velocities (Fig. 7e). In the On SAC model, we found that the "Cell diameter" motion trajectory even produced negative DSI values, indicating a preference for CP motion. Proximal BCs were activated symmetrically in this simulation, while distal BCs favored CP motion due to their position. The overall input was larger during CP motion, resulting in a negative DSI in SACs, which is opposite to experimentally observed motion preference. Importantly, other network mechanisms[57] and SAC intrinsic mechanisms[74,75] have been shown to contribute to the computation of CF motion preference in SACs. Those mechanisms could ensure a preference for CF motion in SACs during motion that does not originate in the SAC center. Thus, BC type-specific surround properties, and by extension their rDS, play a role in establishing directional tuning and spatial RF properties of SACs in our model.

## Starburst amacrine cells respond strongly to motion restricted to short distances

Our SAC dendrite model demonstrates a preference for motion restricted to short distances due to the center-surround interactions at

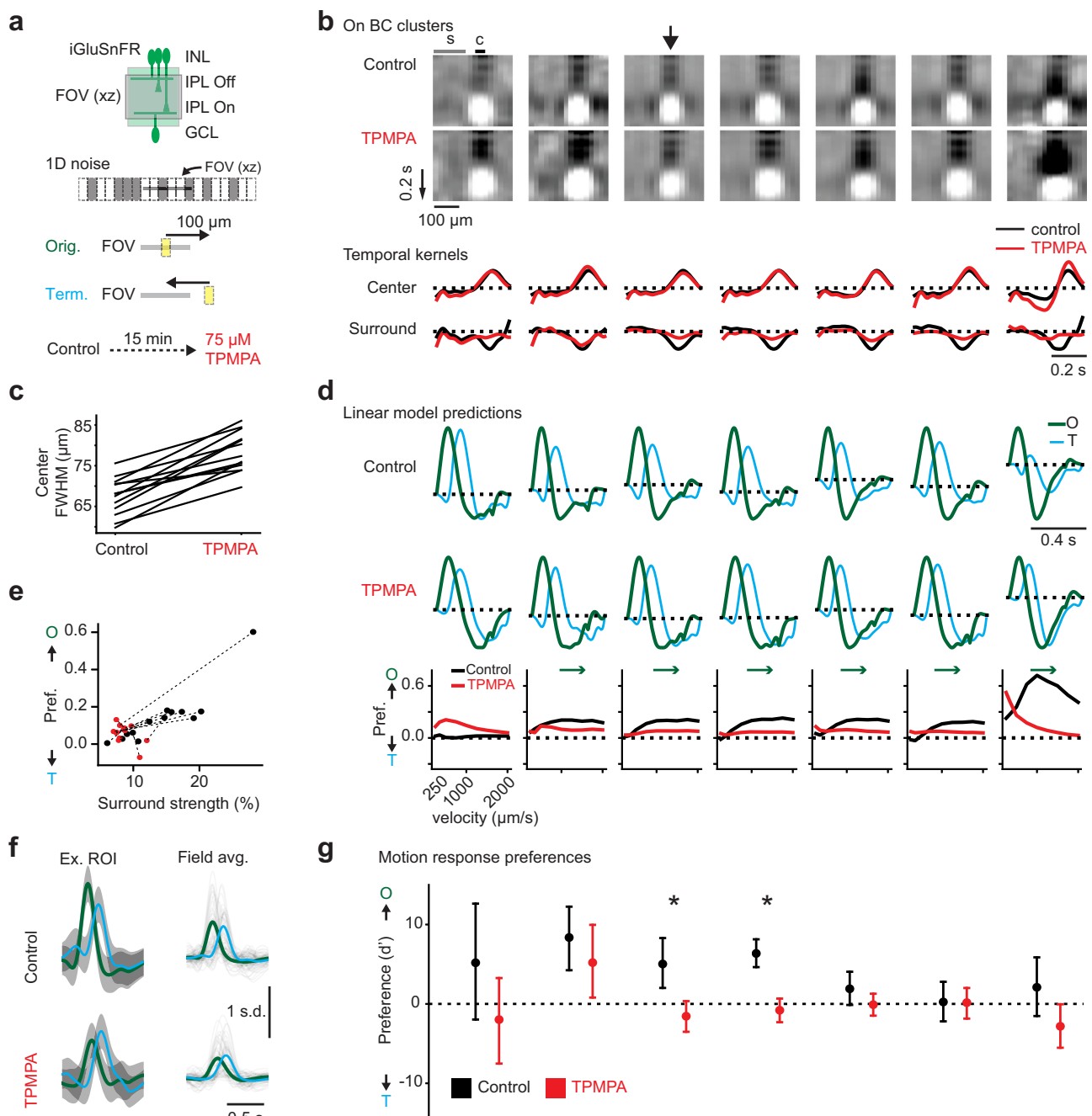

**Fig. 5 | Block of GABA_C receptors weakens radial direction selectivity in two On bipolar cell clusters. a** Top: iGluSnFR imaging. Middle: Moving bar stimulus and "1D noise" stimulus to measure radial direction selectivity (rDS) and assign cluster membership and predict motion responses. Bottom: drug treatment, see Methods for details. **b** Top: average RF of each cluster in control and in TPMPA. Arrow indicates the cluster whose average motion responses are shown in (**f**). Bottom: temporal RFs averaged from center ("c") and surround ("s") regions indicated above, normalized to their respective peaks. Temporal RFs in TPMPA (red) are normalized to the responses in control condition (black). Sample size is 3544 ROIs/ 3 fields/ 3 mice. **c** Full width half maximum (FWHM) of the spatial RF in control and TPMPA. **d** Top: linear convolution model predictions for originating ("O") and terminating ("T") motion using RFs from control or TPMPA conditions. Velocity, 1000 μm/s. Bottom: velocity tuning from these model predictions in control and TPMPA. **e** rDS of each

cluster's modeled responses (at 1000 μm/s) as a function of cluster surround strength. Black dots: control, red dots: TPMPA, dotted lines connect the individual clusters. **f** Left: example response to moving bars (Gaussian Process prediction as in Fig. 1) for a ROI in the cluster indicated with an arrow in (**b**) in control and TPMPA conditions. Colors, estimated mean; gray shading, 3 s.d. Right: individual ROIs' responses (gray) and the cluster average. This was one of two clusters that showed a significant change in the rDS after applying TPMPA (see **g**). **g** rDS (d') for the ROIs in each cluster in control (gray) and TPMPA (red). Points, the mean; error bars, 95% confidence interval. Two-way ANOVA with repeated measures, significant for the effect of TPMPA ($p = 5.87e-11$), but not cluster ($p = 1$) or interaction ($p = 1$). Post-hoc $t$-test with Bonferroni correction, significant for clusters marked with * (from left to right, $p = 0.1387, 0.2398, 0.0017, 6e-10, 0.1329, 0.9587, 0.4649$). Data from 1464 ROIs/ 3 fields/ 3 mice. See also Supplementary Fig. 8.

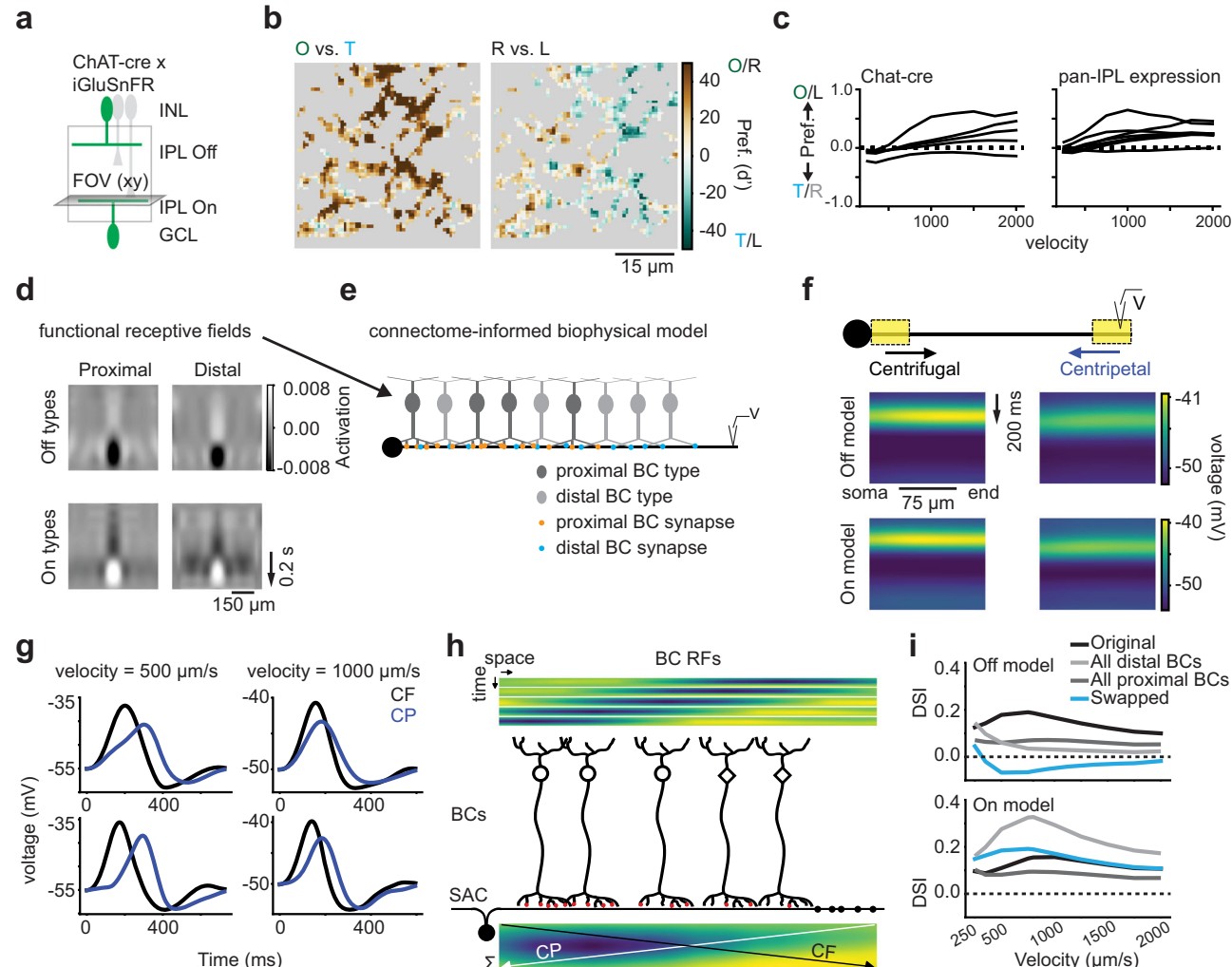

**Fig. 6 | Bipolar cell inputs to starburst amacrine cells are radially direction selective and model amacrine cell inherits bipolar cell motion information.**
**a** flex-iGluSnFR was injected into ChAT-cre mice to achieve SAC-specific labeling of glutamate inputs. **b** The motion preference (d') for all ROIs in an example imaging field showing rDS. "O", originating; "T", terminating; "L", left; "R", right. For details see Supplementary Fig. 9. **c** Comparison of rDS velocity tuning curves for On BC clusters identified from mice expressing iGluSnFR only in the SACs (left, ChAT-cre) vs. ubiquitously (right, from Fig. 2). For details see Supplementary Fig. 10. **d** Up-sampled functional RFs of four BC clusters selected based on co-stratification with SAC-connected BC types. **e** Ball-and-stick multi-compartment model of a single On SAC dendrite with BC inputs organized according to anatomical and physiological data. **f** Membrane potential along the dendrite in an Off (top) and On (bottom) SAC model during centrifugal ("CF") and centripetal ("CP") motion of a moving bar

(20 µm) at 1000 µm/s. **g** Simulated responses of a distal compartment of the two SAC models for two stimulus velocities (500 and 1000 µm/s) during the motion in (**f**). **h** Space-time RFs of individual BCs (top) and their output synapses (red dots) onto a SAC dendrite (black dots: SAC output synapses). The superposition of BC RFs (bottom heatmap) was obtained by summing individual BC RFs, weighted by their number of synapses to the SAC. Proximal BCs are activated by CF motion first and by CP motion last. **i** Direction selectivity index (DSI) of Off and On SAC model distal compartments at different stimulus velocities. We modeled four different BC input distributions. (1) "Original" (black): BC inputs set based on anatomical and physiological data. (2) "All proximal" (dark gray): replaces distal BC inputs with proximal, for one functional type at all input positions. (3) "All distal" (light gray) replaces proximal BC inputs with distal (4) "Swapped" (blue) assigns the proximal BC type to distal locations and vice versa. See also Supplementary Figs. 7, 9 and 10.

the level of the BC inputs (Figs. 6 and 7). We sought to confirm this stimulus preference through RF mapping of the SAC dendrites. We performed 2-photon Ca²⁺ imaging in a mouse expressing the fluorescent Ca²⁺ sensor flex-GCaMP6f or flex-GCaMP8f under the control of the ChAT promoter and presented a noise stimulus to map the RF along one axis (Fig. 8a). We uncovered RFs of small, varicosity-sized ROIs, some of which exhibited a marked motion preference, and clustered these RFs from each scan field into groups using MoG clustering (Fig. 8b). These clusters contained ROIs from areas throughout the FOV, and some clusters appeared to contain ROIs from single dendrites (Fig. 8c). The average RFs from these clusters revealed three patterns: preferring leftward motion, preferring rightward motion and no motion preference (Fig. 8d). These patterns were expected based on the known distribution and outward motion preference of SAC

dendrites in the retina[24]: the population of ROIs exhibiting no motion preference were likely dendrites stimulated off of the axis of their preferred direction, whereas dendrites on the axis of our stimulus exhibited direction preferences. To examine motion preferences across the population, we measured the motion trajectory of each ROI's RF and used this to estimate the preferred motion distance (delta distance) and velocity (Fig. 8e, f). We found that many ROIs did not exhibit a motion preference (delta distance near zero) most likely because these ROIs' dendrites were pointing orthogonal to our stimulus. Among motion-preferring ROIs, the preferred motion travel distance peaked at about 70–100 µm, similar to the size of the SAC excitatory RF radius[32] and the estimated motion distance preference from our model (Fig. 7). We confirmed that SACs respond in a direction selective manner to a stimulus traveling 100 µm by measuring Ca²⁺

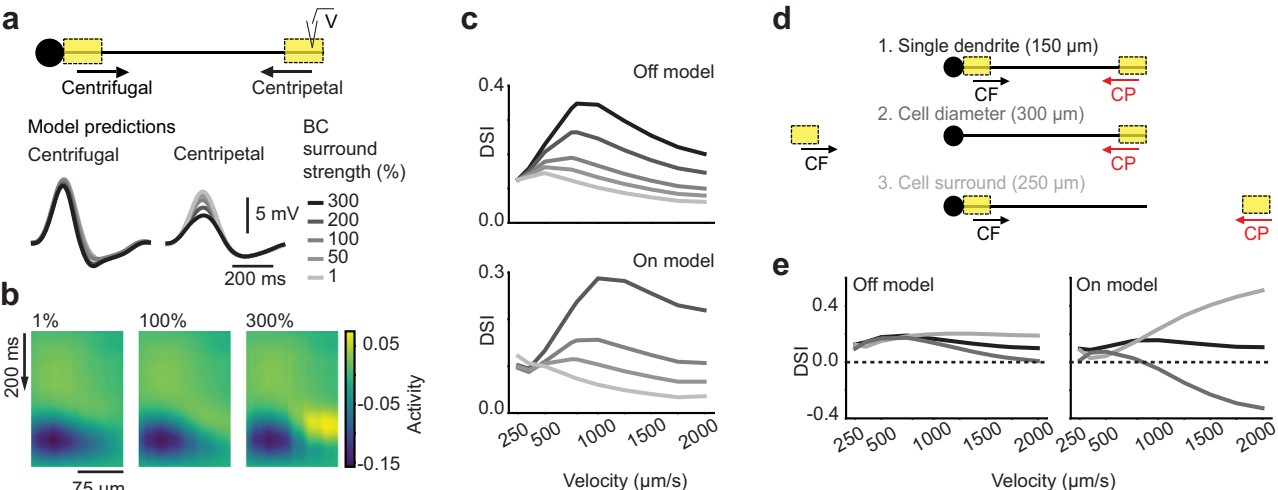

**Fig. 7 | Bipolar cell surround properties shape direction selectivity of model starburst amacrine cell. a** Top: ball-and-stick model starburst amacrine cell (SAC), same as in Fig. 6). Bottom: Predicted responses to a moving bar (1000 μm/s) for SAC models using BC RFs with different surround strengths. **b** Superposition of all BC RFs presynaptic to the Off model at their respective positions along the dendrite at increasing BC surround strengths. **c** Directional tuning (DSI) of the Off and On SAC dendrite models with input from BCs with different RF surround strengths. **d** Spatially different motion stimuli used in the simulations: (1) "Single dendrite": The bar moves along the 150 μm SAC dendrite. (2) "Cell diameter": The motion path includes an additional 150 μm extension to the left, where the other side of the cell would be located. (3) "Cell surround": The motion path includes an additional 100 μm extension to the right of the dendrite, so that the bar moves off the end of the dendrite. **e** DSI of the Off and On SAC models for the different motion stimuli in (m). See also Supplementary Figs. 7 and 10.

responses in their dendrites to moving bars, and found reliable direction selective responses to this stimulus (Fig. 8g–h) with a preference for motion into the FOV (in), as we would expect given the stimulus (Fig. 8g). Thus, SAC RFs and DS are consistent with the rDS of BCs.

## Discussion

In this study, we addressed the question of how BCs in the mouse retina respond to local motion stimulation. We found that some mouse BC axon terminals are sensitive to local motion, responding more strongly to motion originating in their RF center compared to motion originating in their RF surround and traveling to the center, which we call 'radial direction selectivity' (rDS, Fig. 1). Notably, some BCs exhibit rDS, while others do not (Fig. 2), and the level of rDS depends on the BCs' surround strength and timing (Fig. 3a). At every depth of the IPL, at least some terminals exhibited radial direction selective RFs (Fig. 3b–e), suggesting that motion origin information is available to many different circuits. We further studied the properties of BC responses to motion and found that BCs exhibit a preference for motion compared to non-motion stimuli (Fig. 4) and that for two BC clusters rDS depends on GABAergic input (Fig. 5). To determine how BC motion signals are integrated by their postsynaptic partners, we confirmed that SACs receive radially direction selective BC input and modeled how this input is integrated. We found that SACs inherit BC motion signals and this information can be used for direction selective computations (Figs. 6 and 7). We then showed that SAC RFs are shaped by the rDS of BCs (Fig. 8). Our findings suggest that motion signaling arises earlier in the retina than previously thought and that radial direction selective vs. non-selective is an important functional distinction between BC types that informs their contributions to retinal processing.

We found that some types of BCs are capable of signaling information about the origin of moving objects as well as whether objects are looming or receding. Whether or not BCs transmit this information is highly sensitive to the location of the stimulus relative to the BC's RF, as well as the cell's RF properties. Most studies that have previously examined the responses of BCs to moving stimuli did not observe any direction selective tuning in

BC membrane potential[76], intracellular Ca²⁺[43], or glutamate responses[41,42] (but see ref. 45). All of these studies used global motion stimuli, like gratings or wide moving bars, that originated in the RF surround or outside of the RF of the recorded BCs. Under those stimulus conditions, our modeling predicts that BCs respond symmetrically to stimulation (Fig. 2), just as those studies observed. There is one recent study, however, that does not fit into this pattern: using glutamate imaging, the authors provided evidence for a specialized circuit that bestows "true" DS on a subset of axon terminals in type 2 and 7 BCs[45]. Critically, that study found a contribution from wide-field ACs; thus, the mechanism is likely only engaged for large moving stimuli. We found that surround properties were equivalent on two sides of the BC RFs (Fig. 3), suggesting that the BC center-surround motion detector operates symmetrically, conferring rDS. Thus, the motion detectors described here are not cardinally direction selective per se, but they can provide direction information by virtue of their perspective on a motion stimulus.

Our results show that BC terminals have diverse RF properties that may map onto distinct BC types, including striking differences in the strength and temporal properties of the RF surround that contribute to different strengths of rDS (Fig. 3). Many studies have described differences between the RFs of distinct BC types, including differences in the extent and strength of the BC surround[16,43,50]. RF features are tuned by multiple mechanisms in both the outer and inner retina, with differences in dendritic and axonal spread[18], cone inputs[77], horizontal cell influence[78–80], connectivity to ACs[18], inhibitory receptor complement[52,68,81], and susceptibility to neuromodulators[73] all playing a role. Here we found that two putative BC types lose their rDS in the presence of a GABA$_C$ receptor antagonist, suggesting that some BC RF modulation takes place through interactions with ACs (Fig. 5), however other types retained their rDS. Thus, other factors, such as horizontal cell signaling, likely play an important role in establishing the RF properties of at least some BC types. Indeed, a study published in parallel to ours found a strong role for horizontal cell signaling in establishing rDS[82]. One key feature aligned with rDS is a surround that is temporally offset (and slower) than the RF center (Figs. 3 and 4). Thus, it is possible that an initial center-surround structure established

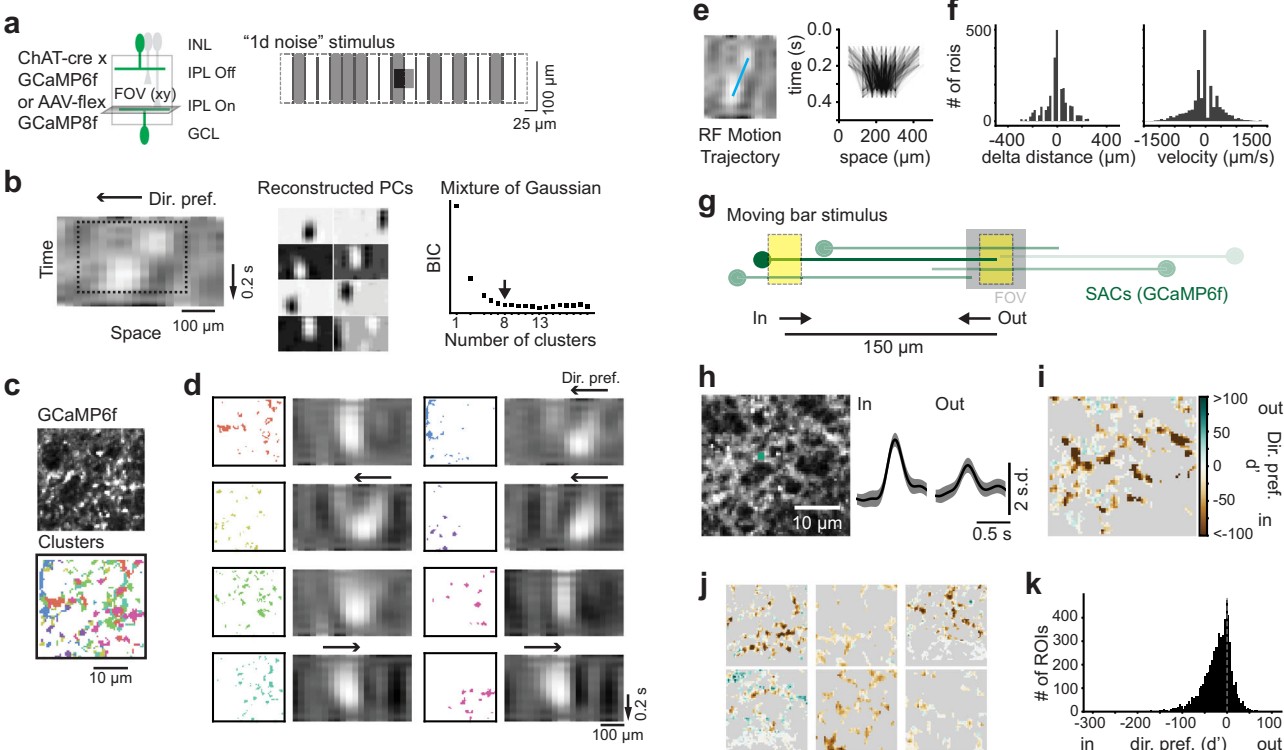

**Fig. 8 | Starburst amacrine cells respond strongly to motion restricted to short distances. a** Left: for starburst amacrine cell (SAC) specific labeling, flex-GCaMP6f and ChAT-cre mice were crossed or ChAT-cre mice were injected with AAV encoding flex-GCaMP8f. Data from both sensors are included. Right: "1D noise" stimulus. **b** Clustering ROIs into groups based on their RFs. Left: example RF with leftward direction preference. Dotted line is cropped region used for clustering. Middle: Reconstructed components of the sparse PCA. Right: Plot of BIC for different number of clusters using Mixture of Gaussian clustering. Arrow: chosen number of clusters (8). **c** Left: example scan field (s.d. image) showing GCaMP6f expression in SACs. Right: ROIs color-coded by clusters determined in (**b**). **d** ROIs within each cluster with their average RFs. **e** Left: example ROI RF showing estimated motion trajectory (blue line). Right: motion trajectories for all ROIs color-coded by cluster ($n = 2164$ ROIs, 5 scan fields, 4 mice). **f** Left: histogram of the change in center position over time (delta distance) for ROIs from (**e**). Right: histogram of the RF velocity measured from line slopes in (**e**). **g** Moving bar stimulus ($20 \times 30$ μm moving at 500 μm/s) traveling in two directions, either into ("in") or out of ("out") the field of view (FOV) and traversing a distance of 150 μm. Positions of different SACs diagrammed over the stimulus (green dendrites and somas), demonstrating that SACs on the left of the FOV will be better stimulated compared to SACs on the right. **h** Left: example scan field (s.d. image) showing GCaMP6f expression and example ROI (green). Right: Gaussian Process prediction for the response to each motion direction for the example ROI. **i** Direction preference (d') estimated from Gaussian Process predictions for all ROIs in example field in (**h**). **j** Additional fields in this dataset showing each ROIs' direction preference. **k** Histogram of d' for all ROIs (6237 ROIs/ 7 fields/ 5 mice). Significant with $p < 0.001$, one sample $t$-test.

in the outer retina is further fine-tuned in the inner retina, utilizing BC type-specific slow AC contributions (i.e., through feedback) to establish strong rDS.

We found that differences in spatio-temporal RF properties are capable of supporting BC feature selectivity with regard to motion stimuli. Importantly, we found this to be the case while modeling the BC responses using linear models based on their measured RFs, which already revealed complex and diverse motion processing across BC clusters (Fig. 2). Increasing evidence suggests that BCs respond in a nonlinear manner to some types of stimuli[36,54,55,63,83–86], and essentially linearly to other types[87]. Our direct observations of motion responses in BCs fit well to our response predictions from linear modeling (Figs. 1, 2 and Supplementary Fig. 4). This could be due to the fact that we model responses based on the full spatio-temporal RFs, which are not space-time separable. In some cases, it is possible that observed nonlinearities in retinal neurons may be the result of either assuming that space and time RFs are separable or taking only space or time components of RFs into account in predictions. It will be interesting to further investigate motion processing in a nonlinear context, which might be particularly important for understanding neuronal responses to natural stimuli. Indeed, the study published in parallel to ours has found that the center-surround RFs of BCs supports novel object detection in natural contexts[82].

Although few studies have observed BC rDS, evidence of this feature of BCs is pervasive in the literature in the form of voltage-clamp recordings of glutamatergic synaptic input to RGCs and ACs. In the mouse retina, the glutamatergic input to both SACs and VGluT3 ACs recorded in response to looming vs. receding stimuli exhibited a strong looming preference[6,31,46], and apparent motion stimuli elicited asymmetric glutamatergic inputs in RGCs[54]. In the primate retina, glutamatergic inputs to several types of RGCs were demonstrated to exhibit motion sensitivity[63,88]. And in the rabbit retina, local apparent motion elicited asymmetric glutamatergic inputs to direction selective RGCs[39]. In some cases, these results have been attributed to voltage clamp errors[89], while in others, they have been attributed to gap junctional interactions between BCs[54,63]. Here, we observed rDS and a preference for motion over random stimuli both experimentally and in models lacking gap junction interactions at the level of BCs. Thus it appears that this preference for motion sequences over random sequences is inherent in the space-time structure of the BC RF, which contains space-time inseparable elements that specifically favor motion stimuli. We propose that, in some cases at least, these many observations of motion sensitivity in downstream cells reflect the rDS we found in subsets of BCs, and that BC RFs play an important role in generating cardinal DS, looming sensitivity, and other types of local motion

sensitivity through the collection of information from rDS BC inputs by diverse downstream neurons.

We explored how BC rDS contributes to one downstream motion computation, DS in SACs. In our measurements of BC glutamate release onto SAC dendrites, we observed clear DS in the input when our stimulus began in many of the model BCs' RF centers. Our modeling suggests that these radial direction selective inputs are integrated to support SAC DS, and are in line with SACs' selectivity for motion towards their dendritic tips[24]. Many studies that have previously evaluated SAC DS used stimuli that activate BC RFs' center-surround rDS detector, including local moving bars[32], differential motion stimuli[90], and expanding rings[24,69,75,91]. We argue that this stimulus dependence is not a bug but a feature of SAC RFs, tuning them to prefer local motion starting close to the SAC soma. Stimulus-dependent motion processing has previously been described in mouse VGluT3-expressing ACs[92], W3 RGCs[70], and rabbit On-Off direction selective RGCs[93], all of which show preferences for local motion. In addition, directional tuning in some On-Off direction selective RGCs in mouse is stronger for local drifting gratings compared to global ones[90] and in rabbit directional tuning in direction selective RGCs is observed for stimuli traveling distances shorter than the spacing between photoreceptors[94]. On the other hand, direction selective RGCs are known to play important roles in brain functions and behaviors involving global motion information[64,95,96]. Previous studies have also suggested that BCs participate in direction detection via other mechanisms (for example see refs. [19], [30], [31], [34], [45], [57], but also refs. [32], [33], [41]) which raises the question of how these mechanisms work together. Given the diverse stimuli often used to probe motion processing, it is possible that distinct mechanisms of direction detection are engaged under different environmental conditions (as previously suggested in refs. [97], [98]), which could ensure robust DS. Thus, studying the role of BC motion signals during local motion processing in SACs and direction selective RGCs could provide important insights to understand the role of BCs in this circuit. Notably, the mechanism of signal suppression during null direction motion that we report here has long been described[53] and has also been observed in the fly visual system (reviewed in ref. [99]) and in the rodent whisker system[100].

Beyond the DS circuit, there are many other RGC types that could rely on BC rDS. In mammals, several RGC and AC types are object motion sensitive, responding specifically to local motion or differential motion[4,6,15,70,76,87,92,101–103], including several prominent primate RGC types, such as parasol RGCs[63,88]. In salamander, a large class of Off ganglion cells prefers motion originating in the RF center compared to motion passing through, which was termed an "alert response to motion onset" and whose responses are best predicted when accounting for the space-time RFs of BCs[56]. In general, RGCs and ACs must fulfill a few requirements to be capable of integrating BC motion information into their computations. The first requirement is that downstream cells receive input from radial direction selective BC types. It will be interesting to explore the wiring of BCs to their post-synaptic partners in this context. The second requirement is that downstream cells should employ post-synaptic integration that allows for motion-related information to be preserved in the cell's output. RGCs are capable of retaining RF structure from BCs[104]; indeed, center-surround interactions at the level of BCs contribute to RGC encoding of spatial features[55,56,84]. A mechanism for local motion integration is hinted at by a recent study that found that the dendrites of some mouse RGC types perform less spatial averaging than others[105]. Since spatial averaging would likely blur spatially-restricted local motion signals (Fig. 1), this integration feature could allow for integration of motion information from BCs. Combining connectomic information about wiring with functional and modeling explorations of RGC and AC responses that take BC RF properties into account, such as we have done here, will thus provide a fruitful avenue for understanding motion processing in the retina.

BC rDS may be relevant in a wide variety of natural conditions important for behavior. Detection of the origin of moving objects and looming detection are highly relevant to animals (reviewed in refs. [106], [107]) and are highly salient to humans[108–110]. In the case of the mouse, they represent prey and predators, respectively[3–6]. The BC radial direction detector is particularly primed to detect moving objects that are initially occluded in a scene, such as a grasshopper jumping out of the grass or a hawk diving from a great distance. At the same time, the BC radial direction detector is rather insensitive to the type of scene motion that occurs when the body, head and eyes smoothly move. This dichotomy allows for detection of behaviorally-relevant moving objects[15]. It is striking that this essential visual information for animal survival is detected already in bipolar cells.

## Methods

### Animal and tissue preparation

All animal procedures were approved by the governmental review board (Regierungspräsidium Tübingen, Baden-Württemberg, Konrad-Adenauer-Str. 20, 72072 Tübingen, Germany) and performed according to the laws governing animal experimentation issued by the German Government. For measuring glutamate release in the IPL, we used either the ChAT-cre transgenic line ($n = 3$; JAX 006410, The Jackson Laboratory[111]) or C57Bl/6 J ($n = 7$, JAX 000664) mice. For $Ca^{2+}$ imaging in SACs, the ChAT-cre transgenic line was crossbred with the Cre-dependent green fluorescent reporter line Ai59D ($n = 3$; JAX 024105[112]) or the ChAT-cre transgenic line ($n = 3$; JAX 006410, The Jackson Laboratory[111]) was injected with a virus for expression of GCaMP8f. We used adult mice greater than 6 weeks old of either sex. Owing to the exploratory nature of our study, we did not use randomization and blinding. No statistical methods were used to predetermine sample size.

Animals were housed under a standard 12 h day/night rhythm at 22° and 55% humidity. For activity recordings, animals were dark-adapted for >1 h, then anesthetized with isoflurane (Baxter) and euthanized by cervical dislocation. The eyes were enucleated and hemisected in carboxygenated (95% $O_2$, 5% $CO_2$) artificial cerebrospinal fluid (ACSF) solution containing (in mM): 125 NaCl, 2.5 KCl, 2 $CaCl_2$, 1 $MgCl_2$, 1.25 $NaH_2PO_4$, 26 $NaHCO_3$, 20 glucose, and 0.5 L-glutamine (pH 7.4). Throughout the experiments, the tissue was continuously perfused with carboxygenated ACSF at ~36 °C, containing ~0.1 μM Sulforhodamine-101 (SR101, Invitrogen) to reveal blood vessels and any damaged cells in the red fluorescence channel[113]. All procedures were carried out under very dim red (>650 nm) light. The positions of the fields relative to the optic nerve were not taken into account in this study. In some cases the retina was cut into pieces and each piece was mounted and imaged separately to prolong the light sensitivity of the tissue.

### Virus injection

For iGluSnFR imaging, we injected the viral constructs AAV2.7m8.h-Syn.iGluSnFR (generated in the Dalkara lab - for details, see ref. [114]; the plasmid construct was provided by J. Marvin and L. Looger (Janelia Research Campus, USA)) or AAV9.CAG.Flex.iGluSnFR.WPRE.SV40 (Penn Vector Core) into C57Bl/6 J and ChAT-cre mouse lines, respectively. For $Ca^{2+}$ imaging in SACs, we injected the viral construct AAV2.7m8.CAG.Flex.GCaMP8f.WPRE.SV40 (Penn Vector Core) into the ChAT-cre mouse line. A volume of 1 μL of the viral construct was injected into the vitreous humor of 4 to 6-week-old mice anesthetized with 10% ketamine (Bela–Pharm GmbH & Co. KG) and 2% xylazine (Rompun, Bayer Vital GmbH) in 0.9% NaCl (Fresenius). For the injections, we used a micromanipulator (World Precision Instruments) and a Hamilton injection system (syringe: 7634-01, needles: 207434, point style 3, length 51 mm, Hamilton Messtechnik GmbH). Imaging experiments were performed 3-4 weeks after injection.

## Two-photon imaging

We used a MOM-type two-photon microscope (designed by W. Denk, MPI, Heidelberg; purchased from Sutter Instruments/Science Products[113]). The system was equipped with a mode-locked Ti:Sapphire laser tuned to 927 nm (MaiTai-HP DeepSee, Newport Spectra-Physics), two fluorescence detection channels for iGluSnFR/GCaMP6f (HQ 510/84, AHF/Chroma) and SR101 (HQ 610/75, AHF), and a water immersion objective (W Plan-Apochromat × 20 /1.0 DIC M27, Zeiss). For image acquisition, we used custom-made software (ScanM by M. Müller and T.E.) running under IGOR Pro 6.37 for Windows (Wavemetrics), taking time-lapsed $64 \times 64$ pixel image scans (at 9.766 Hz) or $128 \times 32$ pixel image scans (at 15.625 Hz). For vertical glutamate imaging in the IPL, we recorded time-lapsed $64 \times 56$ pixel image scans (at 11.16 Hz) using an electrically tunable lens (ETL; for details, see ref. 61).

## Light stimulation

A DLP projector (lightcrafter (LCr), DPM-E4500UVBGMKII, EKB Technologies Ltd) with internal UV and green light-emitting diodes (LEDs) was focused through the objective. The LEDs were band-pass filtered (390/576 Dualband, F59-003, AHF/Chroma), for spectral separation of the mouse M- and S-opsins, and synchronized with the microscope's scan retrace.

In our experiments, photoisomerization rates ranged from -0.5 (black image) to $\sim 20 \times 10^3$ P* per s per cone for M- and S-opsins, respectively (for details, see ref. 115). Two-photon excitation of photopigments caused additional steady illumination of -10$^4$ P* per s per cone (discussed in ref. 113, 116, 117). The center of the light stimulus was adjusted to be on the center of the recording field, and was verified post-hoc either using the receptive fields (RFs) measured from noise or by estimating the location at which the stimulus response onset was fastest for moving bar stimuli. Analysis was adjusted if the stimulus was determined to be off center. For all experiments, the tissue was kept at a constant mean stimulator intensity level for at least 15 s after the laser scanning started and before light stimuli were presented. Stimuli were presented using custom Python software (QDSpy, https://github.com/eulerlab/QDSpy).

Four types of light stimuli were used: (i) small, positive contrast moving bar ($20 \times 40 \, \mu m$ for iGluSnFR, $20 \times 30 \, \mu m$ for GCaMP) appearing in different locations relative to the FOV and moving at 500 μm/s over varying distances (appearance locations, motion directions, and distances specified in Figs. 1, 4, 5, 8, Supplementary Figs. 8 and 9 with 2–3 s between each stimulus presentation; (ii) "1-d noise stimulus" consisting of 20 adjacent rectangles ($20 \times 50 \, \mu m$ for iGluSnFR, $25 \times 100 \, \mu m$ for GCaMP), with each rectangle independently presenting a random black and white (100% contrast) sequence at 20 Hz for 2.5–5.0 s (Figs. 2–5, Supplementary Figs. 8 and 10). (iii) a white looming and receding stimuli consisting of a white spot on black background that appeared and then expanded or retracted at a velocity of 800 μm/s. For looming, the spot started at 10 μm and expanded to 600 μm (Supplementary Fig. 6). (iv) a stimulus to measure "motion vs. random" sequences (Fig. 4, with trials of either apparent motion or random sequences of objects interleaved. This was achieved by presenting 7 rectangles (each rectangle: $20 \times 50 \, \mu m$) in spatial sequence to originate or terminate in the FOV or presenting the 7 rectangles in two distinct random orders. Each rectangle appeared for 33.3 ms, achieving an apparent velocity of 600 μm/s. All stimuli were presented at 100% contrast, were presented in the same pseudo-random order for each imaging field, and were achromatic, with matched photoisomerization rates for mouse M- and S-opsins.

## Pharmacology

TPMPA (Tocris Bioscience cat no. 1040) and strychnine (Sigma-Aldrich cat no. S0532) were used at concentrations of 75 and 0.5 μM, respectively. Each drug solution was carboxygenated before application. For pharmacology experiments, we cut the retina into pieces in order to expose each retina piece to the drug only once. We recorded in control conditions and then bath applied the drugs for 15 min before recording in drug conditions.

## Data analysis

Data analysis was performed using Python 3 and IGOR Pro. Data were organized in a custom-written database schema using DataJoint for Python framework (https://datajoint.io/, version 0.12.8)[118].

## Pre-processing

Pre-processing was performed using custom scripts in IGOR Pro (version 8.0.4.2) and Python 3. First, we measured the s.d. of each pixel and discarded the bottom 50–90% from further analysis. The threshold depended on the experiment: for ubiquitously expressing iGluSnFR, 50–70% of pixels were discarded; for ChAT-cre restricted imaging, 70–90% were discarded because fewer pixels in the imaging field exhibited fluorescence. Traces for each remaining pixel were imported into DataJoint, then high-pass filtered using a Butter filter (0.2 Hz, order = 5) and z-normalized by subtracting each traces' mean and dividing by its s.d. A stimulus time marker embedded in the recorded data served to align each pixel's trace to the visual stimulus with 1.6–2 ms precision. For this, the timing for each pixel relative to the stimulus was corrected for sub-frame time-offsets related to the scanning.

## Radial direction selectivity estimation

To measure average responses of ROIs during low resolution iGluSnFR imaging (Fig. 1c and Supplementary Fig. 6), we drew manual rectangular ROIs at different locations relative to the stimulus position and calculated a binned average of the ROIs' pixels' responses, resampling the response times of each pixel to 63 Hz. This allowed us to resolve higher time resolution than the frame frequency of our imaging and retain the precise alignment to the stimulus timing.

To obtain Gaussian Process (GP) estimates for BC glutamate release and SAC dendritic Ca$^{2+}$, we followed the methods in ref. 58. First, pixel response quality was assessed by calculating the response quality index (as in ref. 16) for each stimulus condition separately. Pixels were discarded if the stimulus condition with the largest quality index value fell below 0.35. Then, ROIs were built automatically from each high quality pixel to include neighboring high quality pixels and to have dimensions around 2 μm (3–9 pixels, average ROI diameter in Fig. 1: $2.29 \pm 0.28 \, \mu m$; in Supplementary Fig. 9: $2.03 \pm 0.34 \, \mu m$; in Fig. 8i–k: $1.69 \pm 0.38 \, \mu m$; in Fig. 5: $1.49 \pm 0.01 \, \mu m$; in Fig. 5: $1.49 \pm 0.01 \, \mu m$; in Supplementary Fig. 8: $1.49 \pm 0.01 \, \mu m$), which is the estimated size of BC boutons[16] and near the resolution limit of our imaging. ROIs were allowed to have some overlap with one another, which improved the signal to noise of our models and made no assumptions about the resolution of our imaging. Because of this, maps of d-prime (d') in Figs. 1d and 8i–j report the measured value only at the center pixel of a ROI. The average response of a ROI's pixels was obtained by resampling the response times at 125 Hz and averaging within time bins.

Then, for each ROI, we created a GP estimate of the response trace using the GPy toolbox (https://sheffieldml.github.io/GPy, version 1.9.9) at 50 Hz, with warping of the time resolution during the period when the moving bar was presented to capture fast response kinetics[58]. For a given ROI, all stimulus conditions were included in the model. We used the Sparse Gaussian Process Regression algorithm with the Radial Basis Function kernel (with parameters with kernel variance = 1.1 and kernel lengthscale = 0.05), and then the model prediction was stored in DataJoint. GP estimates whose mean activity had an s.d. below 0.1 across time for all stimulus conditions were discarded from further analysis, as these regions were considered non-responsive.

d' was estimated for each ROI's GP estimate as follows: the peak response (μ) and the s.d. at this peak (σ) during the time of bar

presentation was measured. For each pair of opposite directions, $d'$ was calculated as:

$$d' = \frac{\mu_1 - \mu_2}{\sqrt{0.5(\sigma_1^2 + \sigma_2^2)}} \quad (1)$$

For each imaging field, the location of the FOV relative to the stimulus was assessed based on RF mapping (see below), if available, or based on the relative response timing of stimuli in opposite directions.

## Receptive field mapping and clustering

RFs were obtained using a modified spike-triggered averaging method that employs a spline basis to estimate smooth RFs (RFEst toolbox: https://github.com/berenslab/RFEst,[62] v.2). First, traces for each pixel and the stimulus trace were up-sampled to the scan line precision (1.6–2 ms) using linear interpolation to align stimuli and responses. The stimulus trace was then mean subtracted so that 50% contrast was set to zero. Then, we formed ROIs using the same method as described for Gaussian Process ROIs (above) except that we did not discard low quality pixels before creating ROIs. ROI diameters in Fig. 2: 2.04 ± 0.07 μm; in Supplementary Fig. 10: 2.69 ± 0.33 μm; in Fig. 8b–f: 1.79 ± 0.21 μm. To restrict ROIs to the IPL in X-Z recordings using the ETL, the border lines of the IPL were manually determined using an s.d. image. Then, we utilized the splineLG function from RFest to obtain the smoothed spike-triggered average for each ROI over a time lag of 0.5 s.

To cluster RFs, we performed sparse principal components analysis (PCA) and mixture of Gaussian (MoG) clustering using libraries and custom scripts in Python as follows: First, we aligned the RF center for each ROI to the same spatial position. Due to the noisy nature of the individual ROI RFs, we accomplished this by first clustering the ROIs within a field using a hierarchical clustering algorithm (SciPy cluster.hierarchy.linkage in Python, https://www.scipy.org, refs. [119], [120], version 1.5.4) and grouping ROIs into clusters using a fixed distance criterion (0.05). This allowed us to obtain average RFs for ROIs with similar RFs within a field, which had the same RF center and polarity. We measured the maximum in these cluster averages (or minimum for Off layer ROIs) and defined this as the RF center for all ROIs in the cluster.

Next, RFs for all ROIs were flattened to one dimension (space-time) and cropped to include the region of the RF that was available for all ROIs. At the precision of our stimulus alignment, it was possible for the stimulus to be off-center of the imaging FOV by up to 100 μm, resulting in a shift of the mapped RFs and, in our data set, an over-representation of one half of the RF (see Fig. 2 and Supplementary Fig. 3). Thus, clustering was performed on just half of the RF. Next sparse principal components (PCs) of the flattened RFs were determined using the sparsePCA function[121] from scikit-learn (https://scikit-learn.org, ref. [122], version 0.21.3). We also determined the depth of each ROI's center in the IPL using the manually-determined IPL boundaries to find the percentage of the IPL thickness at the ROI's center. Together, the RF sparse PCs and IPL depth constituted the feature weights for MoG clustering, which was performed using the scikit-learn mixture.GaussianMixture toolkit[122]. To determine the best number of clusters, we varied the targeted number of clusters between 3 and 19 and estimated the Bayesian information criterion (BIC). We additionally performed an alternative clustering without including IPL depth as a feature (Supplementary Fig. 5). Next, we calculated the average RF for each cluster and estimated the temporal kernels for center and surround in distinct spatial regions from these averages. We defined the center and surround regions by eye and used the same spatial regions across all clusters. We measured several parameters from these temporal kernels: (i) latency was the time between the peak of the center and peak of the surround response; (ii) surround strength was measured as the ratio of the surround peak and center peak; (iii) biphasic index was measured by finding the ratio of the maximum and minimum of the center's temporal kernel; (iv) center full-width half

maximum (FWHM) was determined by calculating the mean spatial kernel during the time of the center response, fitting this to a Gaussian and finding the FWHM of that Gaussian. The clustering procedure was performed separately for each of the data sets in Figs. 2, 8 and Supplementary Fig. 10. Anatomical correlation between the clusters found in Fig. 2 and BC types identified from previously published electron microscopy (EM) reconstructions[18] was performed by obtaining the kernel density estimation using Gaussian kernels (KDE, scipy.stats.gaussian_kde) of the IPL depth of the ROIs in each cluster. These KDE curves were correlated with each BC type from EM to determine the stratification overlap (Fig. 3).

In order to examine the effect of pharmacological manipulations and motion vs. random stimulation on specific BC clusters (Figs. 4 and 5), we collected an additional data set of ROIs that were presented with noise and motion stimuli in both control and drug conditions. We obtained their RFs and predicted their cluster membership in the original MoG model fit from the dataset in Figure 2 using only the RFs in control conditions. Then, we examined the features of the cluster average RFs and motion responses of ROIs from the additional data set for both control and drug conditions to measure how RFs, predicted motion responses, and real motion responses change in the presence of drugs and/or apparent motion stimuli.

To measure the motion trajectory and preferred velocity of SAC RFs (Fig. 8), we found the peak response during a set early and late time window of each ROI's RF and used these two points to determine the delta distance ($x_2 - x_1$) and the velocity (slope: $(x_2 - x_1)/(t_2 - t_1)$).

## Statistical testing

Statistical testing was performed using Python packages pingouin version 0.3.8[123] for Wilcoxon test (Figs. 1 and Supplementary Fig. 9), 2-way repeated measures ANOVA (Figs. 4 and 5), and post-hoc t-tests with Bonferroni correction; we used SciPy's stats package[119] for 1 sample t-test (Fig. 8), Spearman correlation coefficient (Fig. 3), and paired t-test (Supplementary Fig. 2). To test whether the mean correlations between predictions and motion responses in Supplementary Fig. 4 were significant, we performed a permutation test by shuffling the real responses across ROIs and recalculating the correlations and the resulting mean of the population distribution over 1000 iterations. All statistical tests were performed across populations of ROIs.

## Modeling bipolar cell responses from receptive fields

To predict BC responses to moving stimuli from their RFs, we performed convolution between the RFs and stimulus images. The RFs were cropped to center the RF or just contain half of the RF to model responses to different stimulation in space. Convolution was performed at each spatial location of the images independently, and then summed across space to obtain the final temporal predictions of the responses. We validated this convolution modeling method by obtaining predicted responses from the RFs of single ROIs and calculating the Pearson correlation between the predictions and the real motion responses described by the Gaussian Process predictions (as in Fig. 1) for the same ROIs (Supplementary Fig. 4).

We determined a preference index between paired stimuli, including originating vs. terminating motion, left vs. right motion, looming vs. receding motion, motion vs. random, and cardinal DS by measuring the peak response to each stimulus and calculating a preference index:

$$preference = \frac{peak_1 - peak_2}{peak_1 + peak_2} \quad (2)$$

To explore the relationship between motion coherence of random sequences of rectangles and response amplitude predicted from our models, we calculated the motion coherence of each random sequence as an index based on the spatial distance between bars

appearing in sequence.

$$k = \frac{\sum |x_j - x_i| - s_{max}}{s_{min} - s_{max}} \qquad (3)$$

where $x_i$ and $x_j$ are the spatial positions of rectangles in the stimulus adjacent in time, and $s_{min}$ and $s_{max}$ are the maximum and minimum possible spatial sums. This results in a coherence index in which an apparent motion stimulus has $k = 1$ and the least coherent possible stimulus has $k = 0$.

To prepare cluster RFs for use in the SAC biophysical model, we increased the RF resolution in both space and time, denoised the RFs, and used only half of the RF, reflected, to create BC model inputs (Supplementary Fig. 7). To maintain the space-time structure of the RF during interpolation, we performed singular value decomposition (SVD), performed linear interpolation to increase the resolution of the space and time components by 20x and 1.6x, respectively, and then reconstructed the space-time RFs from the first three components. This denoised the RFs while increasing their resolution. Finally, we mirrored the RF to create a full (symmetrical) spatial RF.

To manipulate the strength of the surround in Fig. 7, Supplementary Figs. 6 and 7 we selected values of opposite polarity to the RF center (negative values for On, positive values for Off) in the surround. These values were multiplied by a scalar (0.01, 0.5, 2, 3) to increase or decrease the strength of the surround. For Off RFs, we found that the surround was generally weaker. This could be due to two features of our "1D noise" stimulus: first, that the background on which the row of rectangles was presented was dark, suppressing the surround of Off BCs, and second, that the individual rectangles of the stimulus were small, leading to low total contrast of the stimulus, which has been demonstrated to cause underestimates of surround strength[72]. Thus, we tested the larger increase in surround strength of 300% specifically for Off RFs. In contrast, with 300% surround, On BC cluster surrounds were so strong that resulting model responses were completely suppressed (data not shown).

### Starburst amacrine cell model

To design the anatomical distribution of BC input to the SAC model dendrite, we calculated the number of BC synapses in 10 μm dendritic segments from glutamatergic input labeling in SAC dendrites[32] and assigned them to BC types according to anatomical data about BC type-specific wiring to SAC dendrites[57]. The Off model included anatomical types 1 and 3a, the On model included anatomical type 7 and a generic type 5 (by merging types 5o, 5t and 5i into one). The BCs' RFs were represented by the functional RFs at their respective locations (Figs. 2 and 3). Where multiple BC clusters overlapped, we tested each possible RF cluster (Supplementary Fig. 10). Moving bar stimuli and BC responses were calculated as described above. We included a spontaneous baseline BC activity, which could be regulated up or down by stimulation of the BC center and surround, respectively. BC activity was rectified by clipping values below zero. The BC activation across time became the current injection input to the SAC model dendrite at the respective synapse locations of each BC. The input to each model was scaled such that the maximum depolarization in the most distal model compartment would reach approximately -35 mV at the lowest stimulus velocity. The biophysical SAC ball-and-stick model was implemented in Brian2 version 2.4.2 (https://brian2.readthedocs.io,[124]) running in Python version 3.7.5. The multicompartment model consisted of an iso-potential soma (diameter: 7 μm) and a 150 μm long dendrite. The initial 10 μm of the dendrite had a diameter of 0.4 μm, the remaining dendrite had a diameter of 0.2 μm[32]. In addition to a leak current, the model included Ca²⁺ channels in the distal third of the dendrite[74]. The Ca²⁺ current was translated to a change in the Ca²⁺ concentration via $\gamma_{Ca^{2+}}$ and the Ca²⁺ concentration in each compartment decayed according to an exponential model[125,126] with time

### Table 1 | Model parameters

| Parameter | Value (unit) | Parameter | Value (unit) |
|---|---|---|---|
| $R_i$ | 150 Ω · cm[32] | $E_{Ca^{2+}}$ | 120.0 mV[127] |
| $R_m$ | 21,700 Ω · cm² [32] | $\bar{g}_{Ca^{2+}}$ | 0.013 mS/mm² [127] |
| $C_m$ | 1 μF/cm² [32] | $[Ca^{2+}]_0$ | 50 nM[125] |
| $E_L$ | –54.4 mV[127] | $\tau_{Ca^{2+}}$ | 5 ms[125] |
| $dt$ Brian2 | 0.1 ms | $\gamma_{Ca^{2+}}$ | 20 M/nC |

constant $\tau_{Ca^{2+}}$ (See Table 1). The strength of tuning in the SAC was measured in the distal third of the dendrite. We calculated the DSI from the membrane potential for each compartment in the distal dendrite from the response to centrifugal (CF) and centripetal (CP) motion as

$$DSI = \frac{CF - CP}{CF + CP} \qquad (4)$$

and reported the average of those compartments in the velocity tuning curves (Figs. 6, 7 and Supplementary Fig. 10).

### Reporting summary

Further information on research design is available in the Nature Research Reporting Summary linked to this article.

## Data availability

Data is available from http://retinal-functomics.net/data/.

## Code availability

Custom code is available from our GitHub repository: https://github.com/eulerlab/bc-motion for this paper. Software for generating stimuli is available from https://github.com/eulerlab/QDSpy.

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

## Acknowledgements

We thank Gordon Eske, Merle Harrer, and Zhijian Zhao for excellent technical support, Robert G. Smith and Gregory Schwartz for feedback and discussions. Funding was provided by the German Research Foundation (Heisenberg professorship, BE 5601/8-1, EXC 2064 ML project number 390727645 to PB; SPP 2041 with BE 5601/2-1+2, EU 42/9-1+2 to BE and TE), the German Ministry of Education and Research (Bernstein Award 01GQ1601 to PB), the Bernstein Center for Computational Neuroscience (01GQ1002 to KF), the Tübingen AI Center to PB (01IS18039A), Christiane Nuesslein-Volhard-Stiftung to AV, Universitätsklinikum Tübingen Fortüne Fellowship to AV, the European Research Council (ERC-StG "NeuroVisEco" 677687 to TB), The Wellcome Trust (Investigator Award in Science 220277/Z20/Z to TB), the UKRI (BBSRC, BB/R014817/1 to TB), and the Max Planck Society (M.FE.A.KYBE0004 to KF).

## Author contributions

Conceptualization: S.S., T.B., P.B., T.E., A.V.; Data curation: M.K., A.V.; Formal analysis: S.S., A.V.; Funding acquisition: T.B., P.B., T.E., A.V.; Investigation: M.K., A.V., T.S., K.F.; Methodology: S.S., A.V., K.F., Y.R., T.B., T.E., P.B.; Project administration: P.B., T.E., T.S.; Resources: T.E., P.B.; Software: T.E., A.V., S.S.; Supervision: T.E., P.B., A.V., K.F., Y.R.; Validation: A.V., S.S.; Visualization: A.V., S.S.; Writing - original draft: A.V., S.S., M.K.; Writing - review and editing: all authors.

## Funding

## Competing interests

The authors declare no competing interests.
