## [Peer Review File · Nature Communications]

Center-surround interactions underlie bipolar cell motion sensitivity in the mouse retinaREVIEWER COMMENTS

Reviewer #1 (Remarks to the Author):

The manuscript by Strauss and colleagues explores the spatiotemporal receptive field properties of bipolar cells (BCs) in the mouse retina, focusing on how they affect the encoding of moving stimuli. The authors measure the glutamate output of BCs throughout the retina using iGluSnFR expressed either non-specifically or in starburst amacrine cells (SACs) in Figs. 6-8.

Overall, this is a well-presented and important study. The data and analyses are of high quality. It is comprehensive in its measurements of BC response properties and generally does a good job contextualizing these new measurements with past BC data and using models to explore how they affect motion. The discovery that BCs in either layer of the IPL have a variety of different spatial RF properties (Figs. 3,4) and the measurements and modeling of the influence of BC surrounds on direction selectivity in SAC neurites (Figs. 5-8) are both novel and significant.

This is an exceptionally strong study from the technical perspective, and aside from one small analysis suggestion, I have little to add in that regard. My criticisms and suggestions fall into 2 main categories. First, there are no experiments to probe mechanism even though several (admittedly coarse but useful) pharmacological tools and more specific genetic lines are available. Second, the framing of these responses as indicating “motion sensitivity” could be somewhat misleading to the reader and would benefit from some reframing and additional experiments and analyses that connect to several previous studies of motion sensitivity in the retina and beyond.

1. Minor technical point

The BC ROI clustering method incorporated data both about function and IPL stratification. While this is a reasonable choice (and one used in previous studies by some of these authors), I would like to see in a Supplemental Figure what happens if ROIs are clustered on function alone. It is hard for me to evaluate the relative influence of functional properties and stratification on the clustering result, and I think this would help.

2. Mechanistic experiments

As noted by the authors, BC surrounds can come from horizontal cells in the outer retina and/or amacrine cells (ACs) in the inner retina. Since horizontal cell feedback mostly (or completely depending on who you ask) influences cones, it is unclear (though probably technically possible with some nonlinearities) how it could create surrounds with different strengths and kinetics in different BC types. ACs, of course, are quite diverse, and previous work by some of these authors showed that they diversify BC response kinetics (Franke et al., 2017). Are they also responsible for the diversity of spatial receptive field (RF) properties among the different BC types?

As in the previous study (Franke et al., 2017), pharmacological blockade of glycine and/or GABA receptors would be a useful manipulation to explore the roles of ACs in BC surrounds. As GABAergic ACs are generally assumed to be responsible for spatially extended inhibition, one would predict that GABA receptor blockade should remove BC surrounds while glycine receptor blockade may not. Either way, the results would be informative. GABA_C blockade with TPMPA would also be a nice, specific manipulation since GABA_C receptors are often found on BC terminals (Lukasiewicz et al., 2004) and can have strong effects on surround suppression in excitatory inputs to RGCs (Mani and Schwartz, 2017).

In the outer retina, pH buffering with HEPES has been effective in blocking HC feedback to cones and affecting RGC surrounds in primate (Davenport et al., 2008), and a Cx57 KO animal is available (Shelley et al., 2005), so the authors could re-measure BC RFs and motion sensitivity with HC feedback disrupted.

3. Framing and considerations of other stimuli and gap junctions

While it is underappreciated by many vision scientists, the core result here is not surprising to those

who really understand spatiotemporal RFs. Motion that drives an inhibitory surround before arriving at the excitatory RF center will drive a weaker response than a stimulus that suddenly appears in the RF center and then moves off into the surround. The strength of this effect will depend on the strength of the surround and its timing offset relative to the RF center. A simple linear model captures this effect, and nonlinearities and gain control at the level of BCs and RGCs can predict how this manifests at the level of RGC responses to a variety of different types of motion stimuli (Chen et al., 2014).

The authors use terms like “sensitivity to local motion” and “motion preference” to describe this phenomenon. While these terms are technically correct and applied correctly, the overall notion could be misleading to a reader given that there is an alternative, almost opposite way to frame the same data. BCs with strong (and temporally offset) surrounds could be described as “motion suppressed” because they will respond to flashed stimuli in their RF center much more strongly than to stimuli that hit the RF surround before the center.

An alternative definition of motion sensitivity has been used in other studies of motion processing in BCs (measured at the population level in RGC excitatory input currents rather than glutamate responses from individual BCs). These experiments compared a sequence of bar presentations in motion order (e.g. left to right across the screen) vs. the same set of bars presented in scrambled order (Kuo et al., 2016; Manookin et al., 2018) and summed the total response across the presentation sequence as the response metric. Both these studies found larger responses to “motion order” presentation at the level of RGC excitatory inputs. This stimulus and analysis has also been used downstream to measure motion sensitivity in superior colliculus (Gale and Murphy, 2014).

Motion sensitivity by this definition seems somewhat opposite to the “local motion” sensitivity described in this study. I would expect that BCs with strong RFs would be predicted to be MORE suppressed to the motion order stimulus than to the scrambled order stimulus so they would have negative motion sensitivity by this metric. In a purely linear model, I believe motion sensitivity by this metric is zero, but I may be wrong about that.

In the two retinal studies, the solution to how BCs can have positive motion sensitivity was their electrical coupling along with their output nonlinearity (Kuo et al., 2016; Manookin et al., 2018). While these studies are cited in this manuscript, the authors don't consider this alternate definition of motion sensitivity by performing this experiment, and they don't incorporate electrical coupling between BCs (via All ACs for ON BCs) in their models. An important and unresolved question in this field is whether OFF BCs (which are not coupled via All ACs) can have stronger responses to motion order than random order bar presentations.

References

- Chen, E.Y., Chou, J., Park, J., Schwartz, G., and Berry, M.J. (2014). The neural circuit mechanisms underlying the retinal response to motion reversal. *J. Neurosci.* 34, 15557–15575.
- Davenport, C.M., Detwiler, P.B., and Dacey, D.M. (2008). Effects of pH Buffering on Horizontal and Ganglion Cell Light Responses in Primate Retina: Evidence for the Proton Hypothesis of Surround Formation. *J. Neurosci.* 28, 456–464.
- Franke, K., Berens, P., Schubert, T., Bethge, M., Euler, T., and Baden, T. (2017). Inhibition decorrelates visual feature representations in the inner retina. *Nature* 542, 439–444.
- Gale, S.D., and Murphy, G.J. (2014). Distinct representation and distribution of visual information by specific cell types in mouse superficial superior colliculus. *J Neurosci* 34, 13458–13471.
- Kuo, S.P., Schwartz, G.W., and Rieke, F. (2016). Nonlinear Spatiotemporal Integration by Electrical and Chemical Synapses in the Retina. *Neuron* 90, 320–332.
- Lukasiewicz, P.D., Eggers, E.D., Sagdullaev, B.T., and McCall, M.A. (2004). GABAC receptor-mediated inhibition in the retina. In *Vision Research*, pp. 3289–3296.
- Mani, A., and Schwartz, G.W. (2017). Circuit Mechanisms of a Retinal Ganglion Cell with Stimulus-Dependent Response Latency and Activation Beyond Its Dendrites. *Curr. Biol.* 27, 1–12.
- Manookin, M.B., Patterson, S.S., and Linehan, C.M. (2018). Neural Mechanisms Mediating Motion Sensitivity in Parasol Ganglion Cells of the Primate Retina. *Neuron* 97, 1327-1340.e4.
- Shelley, J.A., Dedek, K., and Weiler, R. (2005). Effects of Connexin57 Deletion on Horizontal Cell

Receptive Field Size in the Mouse Retina. Invest. Ophthalmol. Vis. Sci. 46, 600–600.

Reviewer #2 (Remarks to the Author):

NCOMMS-21-21046, 'Center-surround interactions underlie bipolar cell motion sensing in the mouse retina'.

This study reports on the local motion-evoked response of bipolar cells in the mouse retina. Using two-photon fluorescence glutamate imaging, the authors show that the majority of anatomically/functionally identified bipolar cell types respond more strongly to object motion originating within their receptive field compared with object motion entering their receptive field. The methods used to demonstrate this are cutting-edge. Model simulations show that the direction-dependent difference in bipolar cell response follows from center-surround receptive field interaction: removing the surround makes the difference go away.

The results are of interest, because they show that bipolar cell output in many types is nonlinear in a way that enhances the response to local object motion. I have a major problem, however, with the connection that is made with subsequent claims pertaining to direction selectivity of the bipolar cells themselves and within the BC-SAC-DSGC circuit.

What I take away from the presented results is that bipolar cells are radially direction selective: they have a stronger response for centrifugal compared with centripetal motion. This is distinctly different from direction selectivity in direction selective ganglion cells, which are tuned to 'linear' motion: simply put left-, right-, up- or downward with respect to the retina. To refer to bipolar cell output as direction tuned confuses the radial tuning shown here with the 'linear' tuning of DSGCs.

Many studies have tested if BC output is (linear) direction tuned, and with just one exception (referenced by the authors; #46), there is no evidence for it. So while I think the enhanced motion response is interesting, I do not see data to support the following statements (1-3; below), which in my opinion are misleading and at best will murk the field by confusing radial direction selectivity with linear direction selectivity. The latter is the phenomenon of major focus for explaining direction selective encoding at the level of the retinal ganglion cells. As far as I understand the data presented here, that remains first observed within individual dendritic branches of Starburst amacrine cells, not bipolar cells.

1. Line 13, Abstract: 'motion-sensitive excitation to starburst amacrine cells through direction-specific signals'. Motion-sensitive, yes. But the 'direction specificity' referred to here is radially symmetric – it is not an instance of the direction selectivity seen in the SAC and DSGC.

2. Line 108-111, Introduction: 'Our findings suggest that BCs produce direction selective signals for motion originating in their RF centers, and that these signals can play a role in the computation of local motion direction in SACs.'. Idem.

3. Line 299-301, Results: 'We observed that many of the BC types chosen for model input were direction selective in our simulations, ...'. Idem. The direction selectivity must be qualified in each of these statements. It is RADIALLY direction selective; outward-preferring.

Reviewer #3 (Remarks to the Author):

This manuscript examines the role of bipolar cells in the processing of visual motion in the mouse retina. Bipolar cells play a key role in transmitting the light signal from the photoreceptors to the downstream ganglion cells and amacrine cells. Strauss et al. report that the glutamatergic outputs of some bipolar cells in the mouse retina exhibit motion origin-dependent radial direction discriminability. This effect was further shown to be mediated by the antagonistic, temporally offset center-surround

receptive field (RF) structure of the bipolar cells. Next, the authors used the starburst amacrine cell as an example to illustrate that the local motion sensing property could be passed from bipolar cells to their downstream neurons and thus could in principle contribute to motion processing of the retina.

While this work could be valuable to the field of neuroscience, there is a concern regarding the novelty of some findings. For example, the center-surround RFs of bipolar cell outputs have already been systematically mapped by Franke et al. in 2017 (see Fig 5, and Extended Data Figure 9). There is not an obvious difference between the results from the '1D bar noise' stimulus used to map RFs in the current manuscript and the '1D ring noise' used by Franke et al. Further, the motion origin-dependent local-motion sensing, which is the theme of the current manuscript, could already be predicted by the center-surround RF presented in the earlier work (Franke et al. 2017). What is more, the local motion sensing described in the current study should not generalize broadly to support motion sensing in the way the authors suggest by the title of the manuscript. Indeed, this effect only applies to restricted scenarios when the moving object emerges at the center of a bipolar cell RF. That being said, this effect has been overlooked by many scientists in the field.

The data collection and analysis are performed at a high level. However, another major problem of the manuscript is that the authors mixed the concepts of motion sensing and direction selectivity without unambiguously defining both concepts. Also, the information in many of the figures is redundant and could be compressed significantly. Additionally, some of the key experiments should be repeated in at least three animals, instead of just one or two animals. Specific suggestions are described below in detail.

1. Fig 1 could be removed from the main text. Fig 1b-c shows exemplary iGluSnFR traces evoked by small moving bar stimuli, which are also shown in Fig 2b-c. Fig 1d-e shows exemplary iGluSnFR traces evoked by long moving bar stimuli, as the authors want to illustrate that the origin-dependent motion sensing does not depend on the type of the stimulus (main text line 138). However, in order to reach this conclusion, one would need to sweep through multiple object shapes, only changing the bar length once would not be enough. Therefore, Fig 1b-c, as a control experiment, could either be put into the supplementary materials or else deleted. If it goes into the supplementary materials, the authors should repeat the experiments in at least three mice to determine whether this effect is consistent.

2. The experiments shown in Fig 2 are designed and analyzed to study the effect of motion origin-dependent direction selectivity, instead of motion origin-dependent motion sensing claimed by the authors. In Fig 2b, the authors designed six moving stimuli: rightward-moving bars emerging from different locations (referred as "Out" stimuli in the main text) and terminating at the same location outside the FOV, and leftward-moving bars emerging from the same locations (referred as "In" stimuli in the main text) and terminating at different locations. The stimulus set is complicated and the corresponding neural responses were not appropriately interpreted. The authors defined a motion preference index (d') to quantify the motion sensitivity of the glutamate release. However, d' actually describes the motion direction preference, which would be confusing to readers if interpreted as motion preference. For the '100 μm ' case, d' could be viewed as a radial direction selective index; however for the '150 μm ' and '200 μm ' cases, it is essentially a rightward motion preference index. The authors should be clear about whether they are studying direction selectivity or pure motion sensing. If the latter, the authors should compare the neuronal responses to moving stimuli and corresponding stationary stimuli or compare the iGluSnFR responses to apparent motion stimuli and random bar stimuli (see Manookin et al. 2018, Fig 1 as an example). The authors could further change the origin of the motion stimuli while keeping the end position the same to test whether motion encoding depends on motion origins. Notably, motion encoding of the bipolar cell seems to also depend on the motion end position (e.g.: Fig 2c, RO11: the peak iGluSnFR signal evoked by '100 μm in' stimulus is smaller than that evoked by '150 μm in' or '300 μm in'). All in all, the authors should redesign the experiments and/or reinterpret the previous results. Also, on a side note, the left panel of Fig 2e is a bit confusing, as d' starts to rise before the 0 position.

3. In Fig 3a-h, the authors extract and cluster the linear RFs of bipolar cells in all layers. Next, in Fig

3i-k, the authors used the extracted linear RF model to predict the cell's response to motion stimuli. However, it is unknown whether the linear RF model accurately predicts the glutamate response evoked by moving stimuli. Therefore, the goodness of fit of the model should be shown to the readers. Ideally, the authors would record iGluSnFR signals evoked by 1-D noise and motion stimuli sequentially from the same FOV of bipolar cell terminals, then extract the linear RFs, convolve them with the motion stimuli, and check whether the predictions of the model match the real neuronal responses evoked by motion stimuli. If the model predictions match the real neuronal responses, it will support a causal relationship between the center-surround structure and origin-dependent motion encoding. Also, although the authors labeled center and surround RFs in Fig 3g, it would be beneficial for the readers if the authors could clarify the criteria for selections of center RF and surround RF. On a side note, the aspect ratio of the cropped ROI in Fig 3c does not match that of the dotted region in Fig 3b, which is confusing for this reviewer.

4. Motion sensing and direction selectivity are two related but different concepts. However, they are mixed throughout Fig 5-8. The motion sensitivity is quantified by direction selectivity index (DSI) in Fig 5 and by motion preference index (d') in Fig 6-8. The authors renamed the motion index d' as direction preference index in Fig 8i, k, but emphasized motion responses of SACs in the main text and figure title. The authors should disentangle these two concepts and be consistent in their wording.

5. In Fig 5, the authors should emphasize that the model only captures the real neuronal response when the moving object is restricted to half of the RF. They should explain why the DSI of the 'Cell diameter' case in Fig 5e (bottom left, on model) is negative. This reviewer suspects that this is due to the fact that the model only has half of the SAC dendrites and omits many significant components contributing to the direction selectivity of the SAC, e.g. lateral inhibition (Ding et al. 2016) and distribution of mGluR2 (Koren et al. 2017). The authors should mention that other mechanisms other than center-surround interaction of bipolar cell outputs would contribute to direction selectivity when the moving bar does not emerge from the center of the SAC.

The results presented in Fig. 5i suggest very different effects of perturbing the model bipolar cell RFs for the On and Off circuits converging onto SACs. The manuscript could more clearly explain this discrepancy as it relates to functional differences between On and Off bipolar cell types.

6. Fig 6-7 map the RFs of bipolar cell inputs onto SACs, which is essentially a verification of the layer-specific RFs extracted in Fig 3. As the model in Fig 5 used the RFs extracted in Fig 3 and Fig 8 was framed as the verification of model prediction shown in Fig 5, Fig 6-7 seem to be redundant with earlier figures and should be put into the supplementary materials.

7. In Fig 8, the imaged varicosities in the FOV could come from horizontal SAC dendrites and SAC dendrites of other orientations, e.g. vertical orientation. This could also explain why a good proportion of the ROIs in Fig 8k is not direction-selective. Although the calcium responses in Fig 8 match the model predictions in Fig 5, this is just a positive correlation and should not be viewed as a causal relationship. Also, the experiments shown in Fig 8 should be repeated in at least three animals.

A more direct test of the model would be to perturb the mechanism of the bipolar cell surround and determine how this affects the downstream motion processing in a SAC. The authors seem to favor a mechanism for bipolar cell RF surrounds that depend on amacrine cells, which would explain the diversity of surround properties across the bipolar cell types. If that were the case, the authors could block GABA receptors and determine how this impacts SAC motion processing. However, it has already been shown that blocking GABA receptors does not eliminate direction selectivity in the SAC response (Ding et al. 2016). Therefore, it is unclear what role the bipolar cell surround actually plays in the main function of the SAC.

Minor comments

- on line 38, it sounds like the GluM1 receives an input from photoreceptors.
- on line 73/74, it could be clearer to describe the antagonistic role of the surround. In the case of an On cell, the surround would inhibit the center in response to light.

- on line 108/109, the response cannot be described as direction selective, because when the stimulus starts in the center of a bipolar cell RF, its response moving in every direction is expected to be equal.
- on line 171, it was not clear how the authors determined the n for statistical purposes. Was it the number of ROIs or the number of fields?

References:

Franke, Katrin, et al. "Inhibition decorrelates visual feature representations in the inner retina." *Nature* 542.7642 (2017): 439-444.

Ding, Huayu, et al. "Species-specific wiring for direction selectivity in the mammalian retina." *Nature* 535.7610 (2016): 105-110.

Manookin, Michael B., Sara S. Patterson, and Conor M. Linehan. "Neural mechanisms mediating motion sensitivity in parasol ganglion cells of the primate retina." *Neuron* 97.6 (2018): 1327-1340.

Koren, David, James CR Grove, and Wei Wei. "Cross-compartmental modulation of dendritic signals for retinal direction selectivity." *Neuron* 95.4 (2017): 914-927.

REVIEWER COMMENTS

Reviewer #1 (Remarks to the Author):

The manuscript by Strauss and colleagues explores the spatiotemporal receptive field properties of bipolar cells (BCs) in the mouse retina, focusing on how they affect the encoding of moving stimuli. The authors measure the glutamate output of BCs throughout the retina using iGluSnFR expressed either non-specifically or in starburst amacrine cells (SACs) in Figs. 6-8.

Overall, this is a well-presented and important study. The data and analyses are of high quality. It is comprehensive in its measurements of BC response properties and generally does a good job contextualizing these new measurements with past BC data and using models to explore how they affect motion. The discovery that BCs in either layer of the IPL have a variety of different spatial RF properties (Figs. 3,4) and the measurements and modeling of the influence of BC surrounds on direction selectivity in SAC neurites (Figs. 5-8) are both novel and significant.

This is an exceptionally strong study from the technical perspective, and aside from one small analysis suggestion, I have little to add in that regard. My criticisms and suggestions fall into 2 main categories. First, there are no experiments to probe mechanism even though several (admittedly coarse but useful) pharmacological tools and more specific genetic lines are available. Second, the framing of these responses as indicating “motion sensitivity” could be somewhat misleading to the reader and would benefit from some reframing and additional experiments and analyses that connect to several previous studies of motion sensitivity in the retina and beyond.

We thank the reviewer for their thoughtful and constructive comments on our manuscript. We have added new experimental data and modeling, included additional analyses, and updated our terminology and discussion to address these comments and suggestions. We are convinced that these additions make the manuscript stronger and more interesting to a broader readership.

1. Minor technical point

The BC ROI clustering method incorporated data both about function and IPL stratification. While this is a reasonable choice (and one used in previous studies by some of these authors), I would like to see in a Supplemental Figure what happens if ROIs are clustered on function alone. It is hard for me to evaluate the relative influence of functional properties and stratification on the clustering result, and I think this would help.

We have added Supplemental Figure 6 with new analysis to the manuscript to address this comment. Here, we clustered ROIs without using IPL depth as a feature, which resulted in fewer clusters (7 instead of 13). These clusters are broader with respect to IPL depth. When we compare the ROI membership between the two clustering methods, we find that in some cases including IPL depth as a feature splits a cluster into multiple

clusters, while in other cases clusters are re-mixed. Importantly, when we model motion responses using the average RFs of these new clusters, we still find a diversity of motion responses, with some clusters exhibiting motion origin sensitivity and others not. One interesting exception is the Off BCs, where we find only two clusters, neither of which exhibits much motion origin sensitivity. Thus including IPL depth as a feature uncovers a subset of Off BCs that are sensitive to motion origin.

2. Mechanistic experiments

As noted by the authors, BC surrounds can come from horizontal cells in the outer retina and/or amacrine cells (ACs) in the inner retina. Since horizontal cell feedback mostly (or completely depending on who you ask) influences cones, it is unclear (though probably technically possible with some nonlinearities) how it could create surrounds with different strengths and kinetics in different BC types. ACs, of course, are quite diverse, and previous work by some of these authors showed that they diversify BC response kinetics (Franke et al., 2017). Are they also responsible for the diversity of spatial receptive field (RF) properties among the different BC types?

As in the previous study (Franke et al., 2017), pharmacological blockade of glycine and/or GABA receptors would be a useful manipulation to explore the roles of ACs in BC surrounds. As GABAergic ACs are generally assumed to be responsible for spatially extended inhibition, one would predict that GABA receptor blockade should remove BC surrounds while glycine receptor blockade may not. Either way, the results would be informative. GABA_C blockade with TPMPA would also be a nice, specific manipulation since GABA_C receptors are often found on BC terminals (Lukasiewicz et al., 2004) and can have strong effects on surround suppression in excitatory inputs to RGCs (Mani and Schwartz, 2017).

In the outer retina, pH buffering with HEPES has been effective in blocking HC feedback to cones and affecting RGC surrounds in primate (Davenport et al., 2008), and a Cx57 KO animal is available (Shelley et al., 2005), so the authors could re-measure BC RFs and motion sensitivity with HC feedback disrupted.

We performed new pharmacological experiments to address these comments, and included a new Figure 5, new Supplementary Figure 7, as well as a new results section discussing these experiments. As suggested by the reviewer, we chose to focus on glycinergic signaling and GABA_C receptor-mediated signaling. We found interesting effects of blockade of GABA_C receptors, but not glycine receptors, on motion origin sensitivity in some bipolar cell clusters.

We think that experiments to adequately address the role of horizontal cells in bipolar cell receptive field properties are beyond the scope of this manuscript. While HEPES disrupts some aspects of horizontal cell signalling, there is emerging evidence that horizontal cells also signal directly to bipolar cells by releasing GABA (Newly published research here: (<https://authors.elsevier.com/c/1eF7-3QW8RwqrZ>)). Good methods to address the role of horizontal cells would include reversibly hyperpolarizing them using chemogenetics (as in

Drinnenberg et al. Neuron 2018) and/or genetic manipulations of the individual signaling mechanisms such as knocking out GABA signalling using cre-specific knockout of the vesicular GABA transporter and knocking out the connexin-57 gap junction hemi-channels, as the reviewer suggested. We do not have mice to perform these experiments in-hand at this time, but hope to perform these experiments in a future study.

3. Framing and considerations of other stimuli and gap junctions

While it is underappreciated by many vision scientists, the core result here is not surprising to those who really understand spatiotemporal RFs. Motion that drives an inhibitory surround before arriving at the excitatory RF center will drive a weaker response than a stimulus that suddenly appears in the RF center and then moves off into the surround. The strength of this effect will depend on the strength of the surround and its timing offset relative to the RF center. A simple linear model captures this effect, and nonlinearities and gain control at the level of BCs and RGCs can predict how this manifests at the level of RGC responses to a variety of different types of motion stimuli (Chen et al., 2014).

The authors use terms like “sensitivity to local motion” and “motion preference” to describe this phenomenon. While these terms are technically correct and applied correctly, the overall notion could be misleading to a reader given that there is an alternative, almost opposite way to frame the same data. BCs with strong (and temporally offset) surrounds could be described as “motion suppressed” because they will respond to flashed stimuli in their RF center much more strongly than to stimuli that hit the RF surround before the center.

An alternative definition of motion sensitivity has been used in other studies of motion processing in BCs (measured at the population level in RGC excitatory input currents rather than glutamate responses from individual BCs). These experiments compared a sequence of bar presentations in motion order (e.g. left to right across the screen) vs. the same set of bars presented in scrambled order (Kuo et al., 2016; Manookin et al., 2018) and summed the total response across the presentation sequence as the response metric. Both these studies found larger responses to “motion order” presentation at the level of RGC excitatory inputs. This stimulus and analysis has also been used downstream to measure motion sensitivity in superior colliculus (Gale and Murphy, 2014).

Motion sensitivity by this definition seems somewhat opposite to the “local motion” sensitivity described in this study. I would expect that BCs with strong RFs would be predicted to be MORE suppressed to the motion order stimulus than to the scrambled order stimulus so they would have negative motion sensitivity by this metric. In a purely linear model, I believe motion sensitivity by this metric is zero, but I may be wrong about that.

In the two retinal studies, the solution to how BCs can have positive motion sensitivity was their electrical coupling along with their output nonlinearity (Kuo et al., 2016; Manookin et al., 2018). While these studies are cited in this manuscript, the authors don't consider this alternate definition of motion sensitivity by performing this experiment, and they don't incorporate electrical coupling between BCs (via All ACs for ON BCs) in their models. An important and unresolved question in this field is whether OFF BCs (which are not coupled via All ACs) can have stronger responses to motion order than random order bar presentations.

We have clarified our terminology around motion sensing and now name it “motion origin sensitivity” throughout the manuscript, which we hope indicates that we are specifically referring to sensitivity to motion origin.

Furthermore, we performed additional modeling, experiments and analysis to explore the possibility raised by the reviewer that BCs would respond more strongly to “random order” stimuli compared to “motion order” stimuli. Contrary to the prediction of the reviewer, we found that using both our linear convolution models and in experiments, we observed a preference for motion compared to random stimuli in most BC clusters, even those that do not exhibit a strong preference for the origin vs. the termination of a stimulus. Thus it appears that this preference for motion sequences over random sequences is inherent in the space-time structure of the BC receptive field, which contains space-time inseparable elements that specifically favor motion stimuli. We found this to be a compelling and interesting addition to our study, and have added it to the manuscript as New Figure 4.

As a note, we think that the motion preference observed in New Figure 4 could contribute to the phenomena observed downstream in the studies mentioned by the reviewer (Kuo et al and Manookin et al). In each of these studies, the authors compare apparent motion to random stimuli, but they stimulate across an area with the RGC centered, rather than starting their stimulus in the center, as we did with the BCs in our model and experiments in New Figure 4. But if one examines the area that the authors stimulate over their retinal ganglion cells, they are stimulating with "originating" motion for a subset of the BC mosaic, which could provide input to the downstream RGCs. We would argue that the cell types these authors studied may report the BC input from those BCs. This is something we would like to test in the future.

Here's a diagram where we show these RGC cell types and stimuli to scale, where one can see that some BCs in the dendritic field of the recorded RGCs should be stimulated with an originating motion stimulus.

From Manookin et al. Neuron 2018

From Kuo et al. Neuron 2016

Regarding the suggestion by the reviewer to include an exploration of the role of gap junction coupling in the phenomena we observe here: We agree with the reviewer that this is an important issue, that, along with the role of inhibitory interneurons, needs to be sussed out in detail. However, the proper experiments to understand the role of gap junctions in specific cell types of the retinal circuit would involve the use of cell-type specific gap junction knock-outs, and we believe these experiments are beyond the scope of the current study. In addition, because we observe the motion vs. random preference in New Fig. 4 without the addition of gap junctions to our models, this already suggests that gap junctions are not critical to the motion sensing properties explored here. We have expanded our discussion of this issue in the Discussion at line 618-633:

In some cases, these results have been attributed to voltage clamp errors (Poleg-Polsky 2011), while in others, they have been attributed to gap junctional interactions between BCs (Kuo et al. 2016, Manookin et al. 2018). Here, we observed MOS and a preference for motion over random stimuli both experimentally and in models lacking gap junction interactions at the level of BCs. Thus it appears that this preference for motion sequences over random sequences is inherent in the space-time structure of the BC RF, which contains space-time inseparable elements that specifically favor motion stimuli. We propose that, in some cases at least, these many observations of motion sensitivity in downstream cells reflect the motion origin sensitive responses we found in subsets of BCs, and that BC RFs play an important role in generating DS, looming sensitivity, and other types of local motion sensitivity through the collection of MOS BC inputs in diverse downstream neurons.

References

- Chen, E.Y., Chou, J., Park, J., Schwartz, G., and Berry, M.J. (2014). The neural circuit mechanisms underlying the retinal response to motion reversal. *J. Neurosci.* 34, 15557–15575.
- Davenport, C.M., Detwiler, P.B., and Dacey, D.M. (2008). Effects of pH Buffering on Horizontal and Ganglion Cell Light Responses in Primate Retina: Evidence for the Proton Hypothesis of Surround Formation. *J. Neurosci.* 28, 456–464.
- Franke, K., Berens, P., Schubert, T., Bethge, M., Euler, T., and Baden, T. (2017). Inhibition decorrelates visual feature representations in the inner retina. *Nature* 542, 439–444.
- Gale, S.D., and Murphy, G.J. (2014). Distinct representation and distribution of visual information by specific cell types in mouse superficial superior colliculus. *J Neurosci* 34, 13458–13471.
- Kuo, S.P., Schwartz, G.W., and Rieke, F. (2016). Nonlinear Spatiotemporal Integration by Electrical and Chemical Synapses in the Retina. *Neuron* 90, 320–332.
- Lukasiewicz, P.D., Eggers, E.D., Sagdullaev, B.T., and McCall, M.A. (2004). GABAC receptor-mediated inhibition in the retina. In *Vision Research*, pp. 3289–3296.
- Mani, A., and Schwartz, G.W. (2017). Circuit Mechanisms of a Retinal Ganglion Cell with Stimulus-Dependent Response Latency and Activation Beyond Its Dendrites. *Curr. Biol.* 27, 1–12.
- Manookin, M.B., Patterson, S.S., and Linehan, C.M. (2018). Neural Mechanisms Mediating Motion Sensitivity in Parasol Ganglion Cells of the Primate Retina. *Neuron* 97, 1327-1340.e4.

Shelley, J.A., Dedek, K., and Weiler, R. (2005). Effects of Connexin57 Deletion on Horizontal Cell Receptive Field Size in the Mouse Retina. *Invest. Ophthalmol. Vis. Sci.* 46, 600–600.

Reviewer #2 (Remarks to the Author):

NCOMMS-21-21046, 'Center-surround interactions underlie bipolar cell motion sensing in the mouse retina'.

This study reports on the local motion-evoked response of bipolar cells in the mouse retina. Using two-photon fluorescence glutamate imaging, the authors show that the majority of anatomically/functionally identified bipolar cell types respond more strongly to object motion originating within their receptive field compared with object motion entering their receptive field. The methods used to demonstrate this are cutting-edge. Model simulations show that the direction-dependent difference in bipolar cell response follows from center-surround receptive field interaction: removing the surround makes the difference go away.

The results are of interest, because they show that bipolar cell output in many types is nonlinear in a way that enhances the response to local object motion. I have a major problem, however, with the connection that is made with subsequent claims pertaining to direction selectivity of the bipolar cells themselves and within the BC-SAC-DSGC circuit.

What I take away from the presented results is that bipolar cells are radially direction selective: they have a stronger response for centrifugal compared with centripetal motion. This is distinctly different from direction selectivity in direction selective ganglion cells, which are tuned to 'linear' motion: simply put left-, right-, up- or downward with respect to the retina. To refer to bipolar cell output as direction tuned confuses the radial tuning shown here with the 'linear' tuning of DSGCs.

Many studies have tested if BC output is (linear) direction tuned, and with just one exception (referenced by the authors; #46), there is no evidence for it. So while I think the enhanced motion response is interesting, I do not see data to support the following statements (1-3; below), which in my opinion are misleading and at best will murk the field by confusing radial direction selectivity with linear direction selectivity. The latter is the phenomenon of major focus for explaining direction selective encoding at the level of the retinal ganglion cells. As far as I understand the data presented here, that remains first observed within individual dendritic branches of Starburst amacrine cells, not bipolar cells.

We thank the reviewer for their comments on our manuscript and their careful review of our findings. We have revised our manuscript text to improve the terminology related to different types of motion sensing and included a schematic (New Figure 6H) to clarify the proposed connection between bipolar cell center-surround receptive fields and SAC direction selectivity.

(h) Space-time RFs of individual BCs (top heatmaps) and their output synapses (red dots) onto a SAC dendrite (black dots: SAC output synapses). The superposition of BC RFs (bottom heatmap) was obtained by summing individual BC RF, weighted by their number of synapses to the SAC. Proximal BCs are activated by CF motion first and by CP motion last.

We reference the new schematic in the main text (lines 399-403):

Thus, the anatomical wiring between BCs and the SAC dendrite, together with the BC RF properties, enable stronger activation of BCs along the SAC dendrite during CF motion compared to CP motion in our model (Fig. 6H).

1. Line 13, Abstract: ‘motion-sensitive excitation to starburst amacrine cells through direction-specific signals’. Motion-sensitive, yes. But the ‘direction specificity’ referred to here is radially symmetric – it is not an instance of the direction selectivity seen in the SAC and DSGC.

We agree with the reviewer that this sentence was confusing. This sentence of the abstract now reads:

The models and experiments demonstrated that bipolar cells pass motion origin sensitive excitation to starburst amacrine cells, which helps establish their directional tuning.

2. Line 108-111, Introduction: ‘Our findings suggest that BCs produce direction selective signals for motion originating in their RF centers, and that these signals can play a role in the computation of local motion direction in SACs.’. Idem.

Throughout the manuscript, we have updated our terminology to be more specific about the phenomenon we describe in bipolar cells. What we previously referred to as “motion sensitivity” we now call “motion origin sensitivity”. We have updated the sentence in the introduction with that terminology (lines 108-110):

Our findings suggest that BCs produce motion origin sensitive signals, and that these signals can play a role in the computation of local motion direction in SACs.

3. Line 299-301, Results: 'We observed that many of the BC types chosen for model input were direction selective in our simulations, ...'. Idem. The direction selectivity must be qualified in each of these statements. It is RADIALLY direction selective; outward-preferring.

We have corrected this terminology to be consistent throughout the manuscript, referring to the phenomenon in bipolar cells as "motion origin sensitivity". This sentence was revised to say (line 385-387):

We observed that many of the BC types chosen for model input were motion origin sensitive in our simulations.

In addition, we have clarified in the manuscript more generally that this phenomenon refers to a motion preference that is radially symmetric, revising the results section to include the following (lines 271-278):

Our results indicate that the relative strength and timing of the surround vs. the center are correlated with stronger MOS in BCs. We propose that the BC center-surround RF acts as a motion detector, resembling a Barlow-Levick detector (53), in which spatial and temporal offset of the inhibitory surround and excitatory center establish sensitivity to motion origin (Fig. 4A).

Reviewer #3 (Remarks to the Author):

This manuscript examines the role of bipolar cells in the processing of visual motion in the mouse retina. Bipolar cells play a key role in transmitting the light signal from the photoreceptors to the downstream ganglion cells and amacrine cells. Strauss et al. report that the glutamatergic outputs of some bipolar cells in the mouse retina exhibit motion origin-dependent radial direction discriminability. This effect was further shown to be mediated by the antagonistic, temporally offset center-surround receptive field (RF) structure of the bipolar cells. Next, the authors used the starburst amacrine cell as an example to illustrate that the local motion sensing property could be passed from bipolar cells to their downstream neurons and thus could in principle contribute to motion processing of the retina.

While this work could be valuable to the field of neuroscience, there is a concern regarding the novelty of some findings. For example, the center-surround RFs of bipolar cell outputs have already been systematically mapped by Franke et al. in 2017 (see Fig 5, and Extended Data Figure 9). There is not an obvious difference between the results from the '1D bar noise' stimulus used to map RFs in the current manuscript and the '1D ring noise' used by Franke et al. Further, the motion origin-dependent local-motion sensing, which is the theme of the current manuscript, could already be predicted by the center-surround RF presented in the earlier work (Franke et al. 2017). What is more, the local motion sensing described in the current study should not generalize broadly to support motion sensing in the way the authors suggest by the title of the manuscript. Indeed, this effect only applies to restricted scenarios when the moving object emerges at the center of a bipolar cell RF. That being said, this effect has been overlooked by many scientists in the field.

The data collection and analysis are performed at a high level. However, another major problem of the manuscript is that the authors mixed the concepts of motion sensing and direction selectivity without unambiguously defining both concepts. Also, the information in many of the figures is redundant and could be compressed significantly. Additionally, some of the key experiments should be repeated in at least three animals, instead of just one or two animals. Specific suggestions are described below in detail.

We thank the reviewer for their many helpful suggestions and thorough review of our manuscript. We revised several figures, increased the sample size of key experiments, and performed additional analyses to address these comments.

Regarding the comment about the novelty of the findings in comparison to the work in Franke et al (2017), we think this new dataset and the conclusions are justified and novel for the following reasons:

First, noise and motion responses were tested in the same fields of view in several cases, which was not done in the previous dataset, allowing us to compare predicted and actual motion responses (New Figs. 4, 5, Sup. Fig. 5, Sup. Fig. 7).

Second, the new dataset was collected with improved unbiased methods for sampling the bipolar cell responses in the inner plexiform layer using the electrically tunable lens, which allowed us to capture On and Off BCs in the same field of view.

Third, we would argue that the “1D noise” stimulus allowed us to better capture motion-related properties of the receptive field compared to that work, which used flickering rings. The frequency of the Franke et al. stimulus was relatively high (60 Hz vs. 20 Hz) which, if measuring the ring flickering as an apparent motion stimulus, would result in velocities at the upper extreme of the motion velocity tuning described in the literature.

Fourth, the ring stimulus symmetrically stimulated the RFs, which would not allow one to identify asymmetries in the RF properties. In the end, we did not observe such asymmetries in our dataset.

Fifth, while we agree that this phenomenon may not generalize to all types of motion sensing in the retina, we are convinced that it plays a major role in many previously described motion sensing amacrine and ganglion cells, and that this role should be examined. In many cases, motion origin sensing properties of bipolar cells are underappreciated, and many experimental motion stimuli contain features that will activate the surround properties of bipolar cells, for example, expanding rings, small moving bars, gratings restricted to a small field of view, dot fields, looming/receding stimuli, and differential grating motion. We hope that our contributions from this manuscript will aid those studying retinal responses to these stimuli to incorporate bipolar cell motion origin sensing in their models.

1. Fig 1 could be removed from the main text. Fig 1b-c shows exemplary iGluSnFR traces evoked by small moving bar stimuli, which are also shown in Fig 2b-c. Fig 1d-e shows exemplary iGluSnFR traces evoked by long moving bar stimuli, as the authors want to illustrate that the origin-dependent motion sensing does not depend on the type of the stimulus (main text line 138). However, in order to reach this conclusion, one would need to sweep through multiple object shapes, only changing the bar length once would not be enough. Therefore, Fig 1b-c, as a control experiment, could either be put into the supplementary materials or else deleted. If it goes into the supplementary materials, the authors should repeat the experiments in at least three mice to determine whether this effect is consistent.

We see the reviewer’s point, so we combined the key panels of this old Figure 1 with old Figure 2 to make New Figure 1. In addition, we have updated the text to highlight that these experiments are observational and that they simply played a role in helping us to generate the hypothesis that we test in New Figure 1. The new Results text is below (lines 129-137):

Surprisingly, we observed that the response amplitude in some regions of the FOV depended on the direction of stimuli. Specifically, BCs in areas where motion originated and terminated appeared to release more glutamate when motion

originated nearby. In other regions, the glutamate release appeared symmetric between motion directions.

Based on these observations, we sought to measure whether individual BC terminals exhibit different responses to objects at different points on their motion trajectories.

2. The experiments shown in Fig 2 are designed and analyzed to study the effect of motion origin-dependent direction selectivity, instead of motion origin-dependent motion sensing claimed by the authors. In Fig 2b, the authors designed six moving stimuli: rightward-moving bars emerging from different locations (referred as “Out” stimuli in the main text) and terminating at the same location outside the FOV, and leftward-moving bars emerging from the same locations (referred as “In” stimuli in the main text) and terminating at different locations. The stimulus set is complicated and the corresponding neural responses were not appropriately interpreted. The authors defined a motion preference index (d') to quantify the motion sensitivity of the glutamate release. However, d' actually describes the motion direction preference, which would be confusing to readers if interpreted as motion preference. For the ‘100 μm ’ case, d' could be viewed as a radial direction selective index; however for the ‘150 μm ’ and ‘200 μm ’ cases, it is essentially a rightward motion preference index. The authors should be clear about whether they are studying direction selectivity or pure motion sensing. If the latter, the authors should compare the neuronal responses to moving stimuli and corresponding stationary stimuli or compare the iGluSnFR responses to apparent motion stimuli and random bar stimuli (see Manookin et al. 2018, Fig 1 as an example). The authors could further change the origin of the motion stimuli while keeping the end position the same to test whether motion encoding depends on motion origins. Notably, motion encoding of the bipolar cell seems to also depend on the motion end position (e.g.: Fig 2c, ROI1: the peak iGluSnFR signal evoked by ‘100 μm in’ stimulus is smaller than that evoked by ‘150 μm in’ or ‘300 μm in’). All in all, the authors should redesign the experiments and/or reinterpret the previous results. Also, on a side note, the left panel of Fig 2e is a bit confusing, as d' starts to rise before the 0 position.

These comments have helped us to revise the terminology we use to describe our findings and we think this has improved the manuscript. We have replaced the term “motion sensitive” with “motion origin sensitive” throughout the manuscript. We are avoiding the phrase “direction selectivity” in order to distinguish this from cardinal direction selectivity observed in the direction selective circuit.

In addition, we have revised Figure 2 (now New Figure 1) to clarify the experiment. We now compare between 4 conditions: originating motion, terminating motion, or motion that passes through from the right or the left. We directly compare the amplitudes of the responses in these cases, and compare the relative preference between these in New Figure 1.

We have also included a new experiment in New Figure 4 in which we compare apparent motion to random sequences of rectangles. We find a preference for the origin of motion compared to random stimuli in many bipolar cell clusters.

Regarding the side note about Fig. 2e (now New Figure 1h), the region around the origin of the “originates” stimulus is the spatial region where ROIs exhibit the effect of preferring originating vs. terminating motion. This region of 50 μm is interesting because it is roughly the size of a bipolar cell RF, which implies that BCs encode motion origin at high spatial precision. It should rise before the zero point in this panel.

3. In Fig 3a-h, the authors extract and cluster the linear RFs of bipolar cells in all layers. Next, in Fig 3i-k, the authors used the extracted linear RF model to predict the cell’s response to motion stimuli. However, it is unknown whether the linear RF model accurately predicts the glutamate response evoked by moving stimuli. Therefore, the goodness of fit of the model should be shown to the readers. Ideally, the authors would record iGluSnFR signals evoked by 1-D noise and motion stimuli sequentially from the same FOV of bipolar cell terminals, then extract the linear RFs, convolve them with the motion stimuli, and check whether the predictions of the model match the real neuronal responses evoked by motion stimuli. If the model predictions match the real neuronal responses, it will support a causal relationship between the center-surround structure and origin-dependent motion encoding. Also, although the authors labeled center and surround RFs in Fig 3g, it would be beneficial for the readers if the authors could clarify the criteria for selections of center RF and surround RF. On a side note, the aspect ratio of the cropped ROI in Fig 3c does not match that of the dotted region in Fig 3b, which is confusing for this reviewer.

We thank the reviewer for suggesting this analysis. We believe it strengthens our manuscript. We have added Supplementary Fig. 5 to the manuscript, which evaluates the goodness of fit of our linear convolution model with real motion responses. We find that our modeling method does a good job of predicting responses. For most ROIs, the fit is correlated with Pearson’s $r > 0.7$. These prediction-response pairs are significantly more correlated than shuffled data.

We have updated the Methods to clarify how we defined the center and surround regions of the RFs. The text is reproduced here:

We defined the center and surround regions by eye and used the same spatial regions across all clusters.

The aspect ratio in Fig. 3c (New Fig. 2c) has been adjusted.

4. Motion sensing and direction selectivity are two related but different concepts. However, they are mixed throughout Fig 5-8. The motion sensitivity is quantified by direction selectivity index (DSI) in Fig 5 and by motion preference index (d') in Fig 6-8. The authors renamed the motion index d' as direction preference index in Fig 8i, k, but emphasized motion responses of SACs in

the main text and figure title. The authors should disentangle these two concepts and be consistent in their wording.

We have worked to clarify our terminology throughout the manuscript text. Now when we refer to the phenomenon we study in bipolar cell glutamate release, we use the phrase “motion origin sensitivity” throughout. Only regarding SAC responses, we use the phrase “direction selectivity”.

5. In Fig 5, the authors should emphasize that the model only captures the real neuronal response when the moving object is restricted to half of the RF. They should explain why the DSI of the ‘Cell diameter’ case in Fig 5e (bottom left, on model) is negative. This reviewer suspects that this is due to the fact that the model only has half of the SAC dendrites and omits many significant components contributing to the direction selectivity of the SAC, e.g. lateral inhibition (Ding et al. 2016) and distribution of mGluR2 (Koren et al. 2017). The authors should mention that other mechanisms other than center-surround interaction of bipolar cell outputs would contribute to direction selectivity when the moving bar does not emerge from the center of the SAC.

We have included the following text to explain the negative DSI observed in the On SAC model and reference previously established mechanisms for direction selectivity in SACs (lines 446-457):

In the On SAC model, we found that the ‘Cell diameter’ motion trajectory even produced negative DSI values, indicating a preference for CP motion. Proximal BCs were activated symmetrically in this simulation, while distal BCs favored CP motion due to their position. The overall input was larger during CP motion, resulting in a negative DSI in SACs, which is opposite to experimentally observed motion preference. Importantly, other network mechanisms (56) and SAC intrinsic mechanisms (73, 74) have been shown to contribute to the computation of CF motion preference in SACs. Those mechanisms could ensure a preference for CF motion in SACs during motion that does not originate in the SAC center.

The results presented in Fig. 5i suggest very different effects of perturbing the model bipolar cell RFs for the On and Off circuits converging onto SACs. The manuscript could more clearly explain this discrepancy as it relates to functional differences between On and Off bipolar cell types.

We have included the following explanation (lines 410-417):

Our results suggest that the Off SAC model is more sensitive to changes in the BC distribution, as all alternative distributions resulted in a reduction of the DS compared to the original one. In the On SAC model, on the other hand, the modifications of the BC distributions resulted in differential changes in the DSI. These results reflect the differences in the BC RF properties and anatomical wiring

between the On and Off pathways (Kim et al. 2014, Greene et al. 2016, Ding et al. 2016).

6. Fig 6-7 map the RFs of bipolar cell inputs onto SACs, which is essentially a verification of the layer-specific RFs extracted in Fig 3. As the model in Fig 5 used the RFs extracted in Fig 3 and Fig 8 was framed as the verification of model prediction shown in Fig 5, Fig 6-7 seem to be redundant with earlier figures and should be put into the supplementary materials.

We have moved old Fig 6-7 to Supplementary Figs. 10 and 11 and put the important conclusion panels of these figures into a revised New Figure 6 (parts b,c).

7. In Fig 8, the imaged varicosities in the FOV could come from horizontal SAC dendrites and SAC dendrites of other orientations, e.g. vertical orientation. This could also explain why a good proportion of the ROIs in Fig 8k is not direction-selective. Although the calcium responses in Fig 8 match the model predictions in Fig 5, this is just a positive correlation and should not be viewed as a causal relationship. Also, the experiments shown in Fig 8 should be repeated in at least three animals.

We agree with the reviewer about the conclusion that vertically-oriented dendrites would show up as non-direction selective with our stimulus. We have made this conclusion explicit in the Results (lines 479-484):

These patterns were expected based on the known distribution and outward motion preference of SAC dendrites in the retina (Euler et al. 2002): the population of ROIs exhibiting no motion preference were likely dendrites stimulated off of the axis of their preferred direction, whereas dendrites on the axis of our stimulus exhibited direction preferences.

We agree that our findings in Old Fig. 8 (New Fig. 7) are correlative. We believe that gathering causative data, such as the experiment proposed by the reviewer below, is beyond the scope of the current study because it would require us to know how to fully abolish the BC surround in a type-specific manner - we hope to be able to perform these experiments one day, but don't see this within straightforward reach. We have made the correlative nature of this data explicit in the conclusion of this results section (lines 500-501):

Thus, SAC RFs and direction selectivity are consistent with the motion origin sensitivity of BCs.

We have increased the sample size for the data in Figure 8 (New Figure 7) to include 4-5 mice for all experiments.

A more direct test of the model would be to perturb the mechanism of the bipolar cell surround and determine how this affects the downstream motion processing in a SAC. The authors seem to favor a mechanism for bipolar cell RF surrounds that depend on amacrine cells, which would explain the diversity of surround properties across the bipolar cell types. If that were the case, the authors could block GABA receptors and determine how this impacts SAC motion processing. However, it has already been shown that blocking GABA receptors does not eliminate direction selectivity in the SAC response (Ding et al. 2016). Therefore, it is unclear what role the bipolar cell surround actually plays in the main function of the SAC.

While blocking or knocking out GABA receptors does not eliminate DS in the SAC response, it does strongly impact the strength of the tuning (Lee & Zhou 2006, Chen et al. 2016, Vlasits et al. 2016) including in the experiment the reviewer mentions (Ding et al. 2016). We showed in our model that reducing the strength of the surround strongly limits but does not abolish DS (Fig 6L). As the reviewer points out, other mechanisms, including intrinsic dendritic properties, still tune the dendrite to some extent in this scenario. We argue in the manuscript that bipolar cell motion origin sensing has the ability to strongly tune the dendrite.

One important caveat of the experiment (in Ding et al. 2016) mentioned by the reviewer is that only GABA_A receptors were blocked with SR95531. We performed new experiments using pharmacological blockers and added them as new New Fig. 5 and New Supplementary Fig. 7. We found that some bipolar cell clusters' motion origin sensing is enhanced through other inhibitory signalling mediated by GABA_C receptors. However, not all clusters rely on GABA_C-mediated pathways for their tuning, and more remains to be done to understand mechanisms of establishing the bipolar cell surround.

We believe that without a more targeted disruption of BC surrounds, such as chemogenetic silencing or knockout of key players in establishing the surround properties for specific bipolar cell types, we cannot disentangle the precise role of BCs in SAC DS experimentally. We hope to do these experiments in the future, but feel they are beyond the scope of the current study.

Minor comments

- on line 38, it sounds like the GluM1 receives an input from photoreceptors.

We have removed the mention of GluM1 from this sentence. Thanks for noticing the error.

- on line 73/74, it could be clearer to describe the antagonistic role of the surround. In the case of an On cell, the surround would inhibit the center in response to light.

The sentence now reads (lines 72-74):

For instance, so-called On BCs depolarize and release more glutamate to light increments in the center (a light turning "On"), while the surround antagonistically inhibits the center to light increments.

- on line 108/109, the response cannot be described as direction selective, because when the stimulus starts in the center of a bipolar cell RF, its response moving in every direction is expected to be equal.

The sentence now reads (lines 108-110):

Our findings suggest that BCs produce motion-origin sensitive signals, and that these signals can play a role in the computation of local motion direction in SACs.

- on line 171, it was not clear how the authors determined the n for statistical purposes. Was it the number of ROIs or the number of fields?

The n was the number of ROIs. We have added this to the methods:

All statistical tests were performed across populations of ROIs.

References:

Franke, Katrin, et al. "Inhibition decorrelates visual feature representations in the inner retina." *Nature* 542.7642 (2017): 439-444.

Ding, Huayu, et al. "Species-specific wiring for direction selectivity in the mammalian retina." *Nature* 535.7610 (2016): 105-110.

Manookin, Michael B., Sara S. Patterson, and Conor M. Linehan. "Neural mechanisms mediating motion sensitivity in parasol ganglion cells of the primate retina." *Neuron* 97.6 (2018): 1327-1340.

Koren, David, James CR Grove, and Wei Wei. "Cross-compartmental modulation of dendritic signals for retinal direction selectivity." *Neuron* 95.4 (2017): 914-927.

REVIEWER COMMENTS

Reviewer #1 (Remarks to the Author):

The authors have made substantial improvements in the manuscript based on the reviews.

My remaining comments are minor:

1. The authors should mention the data and model in (Chen et al., 2013), which looks at motion onset sensitivity at the level of RGCs and how it could initiate in BCs. This is very relevant to the findings here.
2. Supplemental figures are cited out of order in the text. They should probably be reordered according to their first citation.
3. Typo on ln 176: "...measuring..." -> "...measured..."

Reference

Chen, E.Y., Marre, O., Fisher, C., Schwartz, G., Levy, J., Silviera, R.A. da, and Berry, M.J. (2013). Alert Response to Motion Onset in the Retina. *J. Neurosci.* 33, 120–132.

Reviewer #2 (Remarks to the Author):

Revised manuscript NCOMMS-21-21046A

The authors have done an excellent job responding to my comments, as well as to those of the other reviewers. Specifically, the distinction between direction selectivity and specific motion selectivity is made carefully and unambiguously now; a recent study (Matsuda et al., *Nature*, Sep2021) with direct relevance to this work and published after the original submission, is appropriately referenced and included in the Discussion.

Introduction gives an excellent, accurate and concise review of the previous work arguing for and against DS tuning at the level of the bipolar cell glutamate output, and the synaptic organization of bipolar cellstarburst amacrine cell connectivity. This is great.

In terms of the results presented by the authors and the interpretation thereof, I have no further concerns. The main takeaway as I see it is that center-surround receptive field interactions in select bipolar cell types enhance the response to local motion. The enhanced response facilitates motion-evoked responses in postsynaptic circuits across the IPL in general, and establishment of direction tuning in starburst amacrine cell-based direction selective circuitry in particular. I think this work is a valuable next step in understanding bipolar cell signaling and retinal motion sensing.

Minor:

Line 177 calls x-z scans 'volumetric', but they really are plane scans just like x-y are, but along another spatial dimension. Since volumetric conjures up the idea of x-y-z volume, consider omitting the term to avoid confusion.

Reviewer #3 (Remarks to the Author):

The authors have addressed several concerns in the previous review, and it is appreciated that it is difficult to satisfy all comments from all reviewers at once. With that said, there are still substantial concerns with terminology and data analysis that the authors should address prior to publication.

1. One main concern is with the terminology 'motion origin sensitivity' (MOS) - adding another term to the field, which seems unclear and unnecessary. It is the strong opinion of this reviewer that the

authors should change the terminology throughout the paper to describe their findings in terms of radial direction selectivity (radial DS). The main reason is that the authors' definition of motion origin sensitivity (MOS) does not match the corresponding experimental results and related analysis and interpretation.

The authors designed four types of stimuli in Figure 1F: Originates (O), terminates (T), right pass through (R), and left pass through (L) and defined MOS accordingly: "We observed that many ROIs exhibit strong glutamate release to motion originating in the FOV, and that ROIs preferred this stimulus to motion that terminated in the FOV or passed through, which we termed "motion origin sensitivity" (MOS) (Fig. 1F)." According to this definition, the authors should show the motion origin sensitive bipolar cell responses follow these patterns: $O > T$, $O > L$ and $O > R$. However, the authors only showed $O > T$ and $L \sim R$ (Figure 1F-J). In fact, the responses of bipolar cell terminals depend on at least two properties: the origin of motion and the termination of motion. Stimuli O vs T (as well as L vs R) have different origins and terminations. The preference (d') for O over T does not necessarily indicate the sensitivity to motion origin alone. It would be more straightforward to interpret $O > T$ as radial DS, as also pointed out by Reviewer 2. Furthermore, in supplementary Figure 2b, the first two ROIs exhibited $O > T$ and $O < R$, which is interesting and *violates* the definition of MOS (as being motion-origin-sensitive requires $O > T$, $O > L$ and $O > R$), yet the authors did not explain this clearly. If the authors focus on radial DS ($O > T$), this contradiction will not be a major logical problem. Therefore, it is strongly suggested that the authors change the terminology—MOS throughout the paper to radial DS.

2. It makes sense to continue using $d'(L \text{ vs } R)$ as a control for radial DS ($d'(O \text{ vs } T)$) to show that bipolar cells do not have non-radial DS. However, the result in Figure 2k raises some major concerns with this control experiment. Indeed, Figure 2k shows that several clusters (2, 6, 7 and 9) have substantial leftward or rightward motion preference that depends on velocity. Presumably this is a completely spurious result of the 'model' for some clusters - which must have an asymmetry in the full receptive field estimate (and this may be the case for the clusters with the noisiest data). If the authors accept this interpretation, it is somewhat troubling - it calls into question whether the other results in this analysis are robust. For example, the leftward/rightward tuning in the model would probably not predict actual DS tuning in experimental results (i.e., there are not likely to be specific leftward detectors among bipolar terminals), which raises questions regarding which aspects of the models can predict actual responses.

3. The framing of the paper could be simplified by presenting the center-surround model near the beginning and making the strong prediction for radial DS, which is an expected property of any center-surround receptive field with a reasonably strong surround that is delayed in time relative to the center. This receptive field structure is already established for many cell types in the retina, including many bipolar cells - as described in the previous Franke et al. paper. It does not seem useful to present the results described in the paper as 'surprising' (in the Abstract and elsewhere), because radial DS is an expected property of a simple center-surround receptive field. With that said, it is also appreciated that no one has done the specific experiment shown here that demonstrates radial DS in bipolar release.

4. The result in Figure 4 does not support the authors' claims regarding motion sensitivity. First, the data testing radial DS (O vs. T) for some reason is very weak in this figure - i.e., most ROIs have a d' near 0. Further, there is a stronger response to O compared to random sequence 2 (R2) but almost no difference from R1. Thus, it appears that taking only two random sequences as control stimuli yielded a systematic difference from outward motion in only one of the two cases. Based on the conclusion in the text, one would expect that O stimuli would be much stronger than T and that O would also be much stronger than any random stimulus (R1 or R2). The data do not support this conclusion. The authors would presumably have to explore many examples of random sequences to understand the results more thoroughly. The strongest conclusion from the current figure seems to be that many cells prefer R2 over R1.

Minor

1. In Figure 2, it would be useful to see the non-normalized surround (like what showed in supplementary Figure 3b) so that the reader could immediately appreciate which clusters have the

weakest surround and would hence be predicted to have the weakest radial DS strength.

2. In Figure 7 (SAC dendritic imaging), the authors should provide evidence that data from GCaMP6f and 8f can be combined, i.e., that data from each sensor yields similar results for this experiment.

3. Line 121: FOV should be an area. '200 um' should be '200 um squared'?

REVIEWER COMMENTS

Reviewer #1 (Remarks to the Author):

The authors have made substantial improvements in the manuscript based on the reviews.

My remaining comments are minor:

1. The authors should mention the data and model in (Chen et al., 2013), which looks at motion onset sensitivity at the level of RGCs and how it could initiate in BCs. This is very relevant to the findings here.
2. Supplemental figures are cited out of order in the text. They should probably be reordered according to their first citation.
3. Typo on In 176: "...measuring..." -> "...measured..."

Reference

Chen, E.Y., Marre, O., Fisher, C., Schwartz, G., Levy, J., Silveira, R.A. da, and Berry, M.J. (2013). Alert Response to Motion Onset in the Retina. *J. Neurosci.* 33, 120–132.

1. We thank the reviewer for reminding us about this very interesting paper. We agree that it is highly relevant and have added a sentence to our discussion highlighting these findings (line 687-692):
In salamander, a large class of Off ganglion cells prefers motion originating in the RF center compared to motion passing through, which was termed an "alert response to motion onset" and whose responses are best predicted when accounting for the space-time RFs of BCs (Chen et al, 2013).
We also added references to this citation at other relevant points in the manuscript.
2. We have re-ordered the supplemental figures.
3. We have corrected the typo.

Reviewer #2 (Remarks to the Author):

Revised manuscript NCOMMS-21-21046A

The authors have done an excellent job responding to my comments, as well as to those of the other reviewers. Specifically, the distinction between direction selectivity and specific motion selectivity is made carefully and unambiguously now; a recent study (Matsuda et al., Nature, Sep2021) with direct relevance to this work and published after the original submission, is appropriately referenced and included in the Discussion.

Introduction gives an excellent, accurate and concise review of the previous work arguing for and against DS tuning at the level of the bipolar cell glutamate output, and the synaptic organization of bipolar cell starburst amacrine cell connectivity. This is great.

In terms of the results presented by the authors and the interpretation thereof, I have no further concerns. The main takeaway as I see it is that center-surround receptive field interactions in select bipolar cell types enhance the response to local motion. The enhanced response facilitates motion-evoked responses in postsynaptic circuits across the IPL in general, and establishment of direction tuning in starburst amacrine cell-based direction selective circuitry in particular. I think this work is a valuable next step in understanding bipolar cell signaling and retinal motion sensing.

We thank the reviewer for the constructive comments and for appreciating our study.

Minor:

Line 177 calls x-z scans 'volumetric', but they really are plane scans just like x-y are, but along another spatial dimension. Since volumetric conjures up the idea of x-y-z volume, consider omitting the term to avoid confusion.

We have removed the word 'volumetric' as suggested by the reviewer.

Please note that in response to a request by reviewer #3, we decided to change the terminology, referring to this case of motion sensitivity now as radial direction selectivity (rDS) instead of motion origin sensitivity (MOS). For this reason, also the title of the manuscript was slightly changed. For details, please see our response to reviewer #3 below.

Reviewer #3 (Remarks to the Author):

The authors have addressed several concerns in the previous review, and it is appreciated that it is difficult to satisfy all comments from all reviewers at once. With that said, there are still substantial concerns with terminology and data analysis that the authors should address prior to publication.

1. One main concern is with the terminology 'motion origin sensitivity' (MOS) - adding another term to the field, which seems unclear and unnecessary. It is the strong opinion of this reviewer that the authors should change the terminology throughout the paper to describe their findings in terms of radial direction selectivity (radial DS). The main reason is that the authors' definition of motion origin sensitivity (MOS) does not match the corresponding experimental results and related analysis and interpretation.

The authors designed four types of stimuli in Figure 1F: Originates (O), terminates (T), right pass through (R), and left pass through (L) and defined MOS accordingly: "We observed that many ROIs exhibit strong glutamate release to motion originating in the FOV, and that ROIs preferred this stimulus to motion that terminated in the FOV or passed through, which we termed "motion origin sensitivity" (MOS) (Fig. 1F)." According to this definition, the authors should show the motion origin sensitive bipolar cell responses follow these patterns: $O > T$, $O > L$ and $O > R$. However, the authors only showed $O > T$ and $L \sim R$ (Figure 1F-J). In fact, the responses of bipolar cell terminals depend on at least two properties: the origin of motion and the termination of motion. Stimuli O vs T (as well as L vs R) have different origins and terminations. The preference (d') for O over T does not necessarily indicate the sensitivity to motion origin alone. It would be more straightforward to interpret $O > T$ as radial DS, as also pointed out by Reviewer 2. Furthermore, in supplementary Figure 2b, the first two ROIs exhibited $O > T$ and $O < R$, which is interesting and *violates* the definition of MOS (as being motion-origin-sensitive requires $O > T$, $O > L$ and $O > R$), yet the authors did not explain this clearly. If the authors focus on radial DS ($O > T$), this contradiction will not be a major logical problem. Therefore, it is strongly suggested that the authors change the terminology—MOS throughout the paper to radial DS.

We thank the reviewer for the detailed explanation of their reasons for suggesting the terminology change. We have decided to follow this advice and use the term 'radial DS' (rDS) throughout the manuscript. Note that this change also affects the title of the manuscript. Because we also demonstrate in Fig. 4 that bipolar cells prefer stimuli with high motion coherence (see response to Major Comment 4 below), we have updated the title to say "Center-surround interactions underlie bipolar cell motion sensitivity in the mouse retina".

In addition, we have taken care to highlight when we are referring to *cardinal* direction selectivity (DS) throughout the manuscript, as a clear delineation of these phenomena was a major criterion of at least one reviewer in a previous version of the manuscript.

2. It makes sense to continue using $d'(L \text{ vs } R)$ as a control for radial DS ($d'(O \text{ vs } T)$) to show that bipolar cells do not have non-radial DS. However, the result in Figure 2k raises some major concerns with this control experiment. Indeed, Figure 2k shows that several clusters (2, 6, 7 and 9) have substantial leftward or rightward motion preference that depends on velocity. Presumably this is a completely spurious result of the 'model' for some clusters - which must have an asymmetry in the full receptive field estimate (and this may be the case for the clusters with the noisiest data). If the authors accept this interpretation, it is somewhat troubling - it calls into question whether the other results in this analysis are robust. For example, the leftward/rightward tuning in the model would probably not predict actual DS tuning in experimental results (i.e., there are not likely to be specific leftward detectors among bipolar terminals), which raises questions regarding which aspects of the models can predict actual responses.

Unfortunately, the data quality is not the same on one side of the cluster average RF compared to the other side because, unintendedly, we sampled less of the RF on one side vs. the other due to slight shifts in stimulus centering (relative to the microscope's recording field). As detailed previously in the Methods, *RFs for all ROIs were flattened to one dimension (space-time) and cropped to include the region of the RF that was available for all ROIs. At the precision of our stimulus alignment, it was possible for the stimulus to be off-center of the imaging FOV by up to 100 μm , resulting in a shift of the mapped RFs and, in our data set, an over-representation of one half of the RF (see Fig. 2 and Supplemental Fig. 3).* Thus, clustering was performed on just half of the RF (see boxed region in Fig. 2b).

For the cluster averages shown in Fig. 2g, we averaged over all available data, which means that the far RF surround often relies on information from fewer ROIs than the near surround and center. We agree with the reviewer that noise in these parts of the RFs is the likely cause of the apparent direction tuning between L vs. R in Fig. 2k. The assumption that the tuning is due to noise is supported by the fact that our dataset of RFs from glutamate imaging on starburst amacrine cells, for which the stimulus was more precisely centered, does not exhibit any tuning to the L vs. R stimulus (Sup. Fig. 10e-f).

We also agree that the tuning to L vs. R in Fig. 2 is misleading. To address this in the revision, we have adjusted the L vs. R model predictions to exclude the noisy far surround (removing 40 μm of the far surround on each side). The model stimulus now traverses 180 μm . Following this adjustment, the tuning curves for L vs. R are now almost completely untuned. We have updated Fig. 2 to display the results of this modeling in panels i-k, as depicted below:

3. The framing of the paper could be simplified by presenting the center-surround model near the beginning and making the strong prediction for radial DS, which is an expected property of any center-surround receptive field with a reasonably strong surround that is delayed in time relative to the center. This receptive field structure is already established for many cell types in the retina, including many bipolar cells - as described in the previous Franke et al. paper. It does not seem useful to present the results described in the paper as 'surprising' (in the Abstract and elsewhere), because radial DS is an expected property of a simple center-surround receptive field. With that said, it is also appreciated that no one has done the specific experiment shown here that demonstrates radial DS in bipolar release.

We have removed the words "surprise"/ "surprisingly" from the abstract, introduction, and results, even though we and many colleagues to whom we have presented this work were genuinely surprised by the pervasiveness of the motion response features found here. We were especially surprised by the fact that radial DS is a feature of only some BCs in our data set. Because there is a dichotomy there, we feel that walking through the evidence in the order we presented it is a useful way to frame our findings. We arrive at the crux of the matter already in Figure 2 and 3. In addition, we feel that we have highlighted the hypothesis that a center-surround RF produces radial DS quite early in the manuscript, that is, already in the introduction (paragraph beginning at line 67).

4. The result in Figure 4 does not support the authors' claims regarding motion sensitivity. First, the data testing radial DS (O vs. T) for some reason is very weak in this figure - i.e., most ROIs have a d' near 0. Further, there is a stronger response to O compared to random sequence 2 (R2) but almost no difference from R1. Thus, it appears that taking only two random sequences

as control stimuli yielded a systematic difference from outward motion in only one of the two cases. Based on the conclusion in the text, one would expect that O stimuli would be much stronger than T and that O would also be much stronger than any random stimulus (R1 or R2). The data do not support this conclusion. The authors would presumably have to explore many examples of random sequences to understand the results more thoroughly. The strongest conclusion from the current figure seems to be that many cells prefer R2 over R1.

To address the reviewer's comments, we tested more random sequences and changed Figure 4 to add these new results and make it clearer.

The reason for radial DS appearing weak in this figure is due to the somewhat low apparent velocity (600 $\mu\text{m/s}$) of the tested stimuli. This velocity was chosen because the framerate of our light stimulator (60 Hz) limits the apparent velocity for small distances. We designed the stimulus to dwell for 2 frames at each spatial position. At the 60 Hz refresh rate, two consecutive stimuli spaced 20 μm apart, the apparent velocity is $20 \mu\text{m} / 0.03334 \text{ s} \approx 600 \mu\text{m/s}$. These details were already listed in the Methods and we have now also added them to the figure legend for further clarity. In addition, we have plotted the predicted tuning from our simulations alongside the actual experimental tuning to show that the model predictions correlate with the experimental findings (Fig. 4j, dotted lines).

As the reviewer noticed, the predicted and actual responses to the two random sequences R1 and R2 differed from one another, and only R1 strongly differed from O and T. Based on the reviewer's suggestion, we explored additional random sequences in our modeling to try to understand the relationship between the predicted response amplitudes and the stimuli. We found that there was a relationship between the extent of motion coherence in the random sequence and the response amplitudes, so we plotted the relationship between a measure of motion coherence and the maximum amplitude of the predicted response. This new analysis is shown in a new panel, Fig. 4h. Based on this analysis, the difference between the responses to R1 and R2 is related to the fact that R2 has more motion coherence than R1. This matches our experimental findings (in Fig. 4i-j) that R1 produced much smaller amplitude responses than the other stimuli, and supports our conclusion that bipolar cells prefer motion to non-motion stimuli.

Minor

1. In Figure 2, it would be useful to see the non-normalized surround (like what showed in supplementary Figure 3b) so that the reader could immediately appreciate which clusters have the weakest surround and would hence be predicted to have the weakest radial DS strength.

We have updated Fig. 2h to display the surround normalized to the center rather than normalized to its own peak, thus showing the relative strength of the surround.

2. In Figure 7 (SAC dendritic imaging), the authors should provide evidence that data from GCaMP6f and 8f can be combined, i.e., that data from each sensor yields similar results for this experiment.

As requested, here are the stacked histograms of the population data from Fig. 7 showing the results by indicator. There are indeed slight differences between the data for the two indicators, but our main point can be made with either data set.

3. Line 121: FOV should be an area. '200 um' should be '200 um squared'?

Yes, 200 um squared is correct. We have corrected this.

REVIEWER COMMENTS

Reviewer #3 (Remarks to the Author):

The authors have made many improvements to the manuscript. Nice work.

A minor comment: the schematics in Fig. 4e look like they are mislabeled. 'Random 1' looks more coherent but has a lower k value than 'Random 2'.

Response to reviewers

REVIEWERS' COMMENTS

Reviewer #3 (Remarks to the Author):

The authors have made many improvements to the manuscript. Nice work.

A minor comment: the schematics in Fig. 4e look like they are mislabeled. 'Random 1' looks more coherent but has a lower k value than 'Random 2'.

We thank the reviewer for their time and interest in our work. The review process has improved the manuscript.

Regarding the minor comment, we have double checked the coherence calculations. According to the way we scored the stimuli, the reviewer is incorrect and the figure is correct: 'Random 1' is less coherent than 'Random 2'. We include a small figure showing how we calculated the coherence score for each of these stimuli. The intuition here is that there are larger "jumps" in spatial position for temporally adjacent squares in the stimulus for 'Random 1' than for 'Random 2', resulting in a higher score and lower index value.

As a result of double checking our coherence index calculations, we did uncover a typo in how these coherence scores were normalized for the coherence index. The values in the figure did not correspond to the coherence equation printed in the methods because scores were divided by $(s_{\min} - s_{\max} - 1)$ rather than $(s_{\min} - s_{\max})$. This error does not change

the results in terms of the relative coherence indices of different stimuli, but it does change the absolute coherence index values. We have corrected this and we include here both the originally submitted figure and the corrected figure, in which the printed coherence values are updated in panels e and h as well as the plots in panel h.

Corrected:

Original: